# Rethinking PCA Through Duality

**Jan Quan**[*]
ESAT-STADIUS & Leuven.AI
KU Leuven, Belgium
jan.quan@kuleuven.be

**Johan Suykens**
ESAT-STADIUS & Leuven.AI
KU Leuven, Belgium
johan.suykens@kuleuven.be

**Panagiotis Patrinos**
ESAT-STADIUS & Leuven.AI
KU Leuven, Belgium
panos.patrinos@kuleuven.be

## Abstract

Motivated by the recently shown connection between self-attention and (kernel) principal component analysis (PCA), we revisit the fundamentals of PCA. Using the difference-of-convex (DC) framework, we present several novel formulations and provide new theoretical insights. In particular, we show the kernelizability and out-of-sample applicability for a PCA-like family of problems. Moreover, we uncover that simultaneous iteration, which is connected to the classical QR algorithm, is an instance of the difference-of-convex algorithm (DCA), offering an optimization perspective on this longstanding method. Further, we describe new algorithms for PCA and empirically compare them with state-of-the-art methods. Lastly, we introduce a kernelizable dual formulation for a robust variant of PCA that minimizes the $l_1$-deviation of the reconstruction errors.

## 1 Introduction

Principal component analysis (PCA) is one of the most fundamental dimensionality reduction techniques and has seen widespread use in various fields, including machine learning [20, 10, 38, 23], signal processing [4, 24], and computer vision [52, 78, 82], due to its ability to efficiently extract meaningful features from high-dimensional data. Its nonlinear extension, kernel PCA [51], extends this method through a feature mapping into a high-dimensional space (possibly infinite-dimensional), enabling the discovery of nonlinear patterns. Crucially, this is achieved without ever computing the transformed data explicitly, using the kernel trick [11].

In recent years, (kernel) PCA has regained interest due to its connection to the self-attention mechanism [79], which lies at the heart of the widely successful transformer architecture. Transformers have revolutionized a wide range of applications such as natural language processing [13, 6, 42], computer vision [25, 59, 49, 76], and speech processing [41, 50]. This success has led to many attempts to understand the underlying mechanisms from a more principled viewpoint.

In particular, [69] establishes a direct link beween self-attention and kernel PCA by showing that the attention outputs are projections of the query vectors onto the principal components axes of the key matrix in a feature space. In a similar vein, [18] recovers self-attention from an asymmetric kernel singular value decomposition [67], which can be seen as a generalization of kernel PCA to multiple data sources. Moreover, it has also been shown that models based on low-rank matrix decompositions can achieve results on par with modern transformers when trained correctly [32].

---

[*]Corresponding author.

39th Conference on Neural Information Processing Systems (NeurIPS 2025).

Motivated by these modern connections to PCA, we aim to revisit the fundamentals of PCA. Our main tool is the concept of difference-of-convex (DC) duality, which provides a general framework for analyzing nonconvex problems, as it is for example known that any $C^2$ function admits a difference-of-convex formulation [34]. One parameter-free algorithm that is suited for solving these types of problems is the difference-of-convex algorithm (DCA) [68], of which the convex-concave procedure [81] is a special case. This algorithm, which fits into the majorization-minimization framework [66], has found many successful applications in machine learning including expectation-maximization [21], successive linear approximation [12], and various clustering methods [3, 56].

**Contributions**   Our contribution is threefold.

1. We derive three novel DC dual pairs for PCA. For these formulations, we compare some simple gradient methods with existing solvers. Moreover, we show that if one of the functions in the primal is unitarily invariant, its DC dual is kernelizable and out-of-sample applicable.

2. We show that simultaneous iteration [77, Algorithm 28.3] is an instance of DCA applied to the variance maximization objective, which in turn is related to the famous QR algorithm for computing eigenvalues of dense matrices. This result gives a new connection between numerical linear algebra and optimization, akin to the connection between (linear) conjugate gradients [35] and accelerated gradient descent.

3. Based on a least absolute deviation formulation for the robust subspace recovery problem, we provide a novel DC formulation that is kernelizable. Moreover, we show that DCA is related to approaches based on iteratively reweighted least squares for this problem.

**Related work**   The idea of using DC duality on the variance maximization formulation of PCA is not new. In particular, it has been studied for one component in [9, 36, 5] while more recent work extended it to multiple components [74]. Nevertheless, we propose three DC pairs that are based on other formulations of PCA and have, to the best of our knowledge, not been considered before.

The QR algorithm [31] is a classical algorithm for computing the eigenvalues of dense matrices. It is known as one of the top ten algorithms of the 20th century [19] and still the de facto industry standard in high-performance computing libraries and software such as MATLAB. The connection between the power method without normalization and DCA is described in [71, Prop. 4]. Notably, this connection is only made for the leading principal component. In contrast, we show that DCA applied to the variance maximization objective of PCA yields a fundamental link with simultaneous iteration [77, Algorithm 28.3], which in turn is related to the QR algorithm. Some more advanced algorithms for solving the same formulation can be found in [2, 64, 1, 27].

Beyond standard PCA, there has been significant interest in incorporating additional structure such as sparsity, or enhancing robustness to noise and outliers [15, 53, 72, 40, 28, 29, 71, 48]. In the kernel setting, [74] extends DC duality to robust PCA by considering other variance-like objectives, though this complicates interpretation and the out-of-sample extension was not considered.

## 2   Preliminaries

### 2.1   Notation

We denote by $\langle \cdot, \cdot \rangle$ the Euclidean inner product for vectors and the Frobenius inner product for matrices. Let $T \in \mathbb{R}^{m \times n}$. We denote by $T_{i,:}$ the $i$th row of $T$. The Schatten $p$-norm of $T$ with $p \in [1, +\infty)$ is defined through $\|T\|_{S_p}^p := \sum_{i=1}^{\min(m,n)} (\sigma_i(T))^p$ where $\sigma(T) \in \mathbb{R}^{\min(m,n)}$ denotes the vector of singular values of $T$ in nonincreasing order. $S_\infty$ is defined as the spectral norm, i.e., the largest singular value of its argument. The Schatten 1-norm is also known as the nuclear norm or trace norm. The Schatten 2-norm is also known as the Frobenius norm or the Hilbert–Schmidt norm. A (full) SVD of $T$ is denoted as $U \operatorname{Diag}(\sigma(T)) V^\top$, where $\operatorname{Diag}(\sigma(T)) \in \mathbb{R}^{m \times n}$ is a rectangular diagonal matrix and $U, V$ are real orthogonal. If $T$ is square, $\lambda(T)$ denotes the eigenvalues of $T$ in any order. We use $\overline{\operatorname{Diag}(\lambda(T))}$ to denote a square diagonal matrix with $\lambda(T)$ on the diagonal. The (closed) unit ball of a norm $\|\cdot\|$ on $\mathcal{X}$ is denoted by $B_{\|\cdot\|} := \{x \in \mathcal{X} \mid \|x\| \leq 1\}$. The extended reals are denoted by $\overline{\mathbb{R}} := \mathbb{R} \cup \{\pm\infty\}$. Let $f : \mathbb{R}^n \to \overline{\mathbb{R}}$. The convex subdifferential of $f$ is denoted by

$\partial f$, whereas its convex conjugate is denoted by $f^*$. The function $f$ is called absolutely symmetric if $f(\gamma) = f(\hat{\gamma})$ for all $\gamma$ and where $\hat{\gamma}$ denotes the vector with components $|\gamma_i|$ in decreasing order. A function $F : \mathbb{R}^{m \times n} \to \overline{\mathbb{R}}$ is unitarily invariant if $F(VXU) = F(X)$ for all $X \in \mathbb{R}^{m \times n}$ and both $U \in \mathbb{R}^{n \times n}, V \in \mathbb{R}^{m \times m}$ are real orthogonal.

**Remark 2.1.** For clarity of exposition, we perform our complete discussion in Euclidean spaces. The extension to infinite-dimensional Hilbert spaces for kernel methods only needs some additional technical details. More concretely, let $\phi : \mathbb{R}^d \to \mathcal{H}$ be some feature mapping and $\{x_i\}_{i=1}^N \subset \mathbb{R}^d$ the given data. Any time a formulation contains the data matrix $X = (x_1^\top \cdots x_N^\top)^\top$, it can be formally replaced by $\Gamma : \mathcal{H} \to \mathbb{R}^N$, where $(\Gamma w)_i = \langle \phi(x_i), w \rangle_\mathcal{H}$ for all $w \in \mathcal{H}$, and $\langle \cdot, \cdot \rangle_\mathcal{H}$ denotes the inner product of the Hilbert space. The kernel matrix is then $K := \Gamma\Gamma^* = [\langle \phi(x_i), \phi(x_j) \rangle_\mathcal{H}]_{i,j=1}^N$. We also employ the following informal definition to denote formulations which allow for practical implementations.

**Definition 2.2** ((Informal) kernelizability)**.** *Let $X \in \mathbb{R}^{N \times d}$ be the data matrix. A problem formulation is said to be **kernelizable** if it can be written solely in terms of the kernel matrix $K = XX^\top \in \mathbb{R}^{N \times N}$. If a problem is kernelizable, we moreover call it **out-of-sample applicable** if for a new data point $\tilde{x} \in \mathbb{R}^d$, the 'output' of the problem can be computed from only $K$ and $X\tilde{x}$.*

## 2.2 Difference-of-convex duality

Our primary tool for deriving new formulations is difference-of-convex (DC) duality, also known as Toland duality, which yields a pair of primal-dual problems between which strong duality holds.

**Proposition 2.3** (Toland duality [73])**.** *Let $G : \mathbb{R}^{d \times s} \to \overline{\mathbb{R}}$, $F : \mathbb{R}^{N \times s} \to \overline{\mathbb{R}}$, be two convex, closed and proper functions, and $X : \mathbb{R}^{d \times s} \to \mathbb{R}^{N \times s}$ a linear mapping. Then, the following pair of primal-dual problems*

$$\underset{W \in \mathbb{R}^{d \times s}}{\text{minimize}} \ G(W) - F(XW) \tag{P}$$

$$\underset{H \in \mathbb{R}^{N \times s}}{\text{minimize}} \ F^*(H) - G^*(X^\top H) \tag{D}$$

*have the same infimum, i.e., **strong duality** holds. If $W^\star$ is a solution of (P), then any $H^\star \in \partial F(XW^\star)$ is a solution of (D). If $H^\star$ is a solution of (D), then any $W^\star \in \partial G^*(X^\top H^\star)$ is a solution of (P).*

Since this DC duality is a special case of the duality described in [62, 11H], the way the primal problem is perturbed/modified can lead to vastly different dual problems, as will become clear from comparing DC dual pairs (l)-(m) and (n)-(o) in Figure 2.

## 2.3 Difference-of-convex algorithm

An effective and widely used algorithm to solve DC problems of the form (P) is the difference-of-convex algorithm (DCA) [68]. The iterates of DCA are given by

$$H^{(k)} \in \partial F(XW^{(k)}); \quad W^{(k+1)} \in \partial G^*(X^\top H^{(k)}) = \underset{\overline{W} \in \mathbb{R}^{d \times s}}{\arg\min} \ G(\overline{W}) - \langle \overline{W}, X^\top H^{(k)} \rangle. \tag{DCA}$$

To make sure the iterates are well-defined, we also require the following constraint qualification.

**Assumption 2.4.** $X \operatorname{dom}(\partial G) \subseteq \operatorname{dom}(\partial F)$ and $X^\top \operatorname{range}(\partial F) \subseteq \operatorname{range}(\partial G)$.

We have the following relation between the iterates of applying (DCA) to (P) and (D) respectively.

**Proposition 2.5.** *Let $\{W^{(k)}\}_{k=0}^\infty$ be the sequence of iterates obtained from applying (DCA) to (P) starting from some $W^{(0)} \in \mathbb{R}^{d \times s}$. Then, there exists a sequence of iterates $\{\tilde{H}^{(k)}\}_{k=0}^\infty$ obtained from applying (DCA) to (D) starting from $\tilde{H}^{(0)} \in \partial F(XW^{(0)})$ such that $W^{(k+1)} \in \partial G^*(X^\top \tilde{H}^{(k)})$.*

The convergence properties for (DCA) are well-established in the literature [68, 63, 55]. We summarize the basic results for completeness. Consider the problem (P) and suppose that the iterates $\{W^{(k)}\}_{k=0}^\infty$ are generated by (DCA).

- Since (DCA) fits into the majorization-minimization framework [66], it is a descent method (without linesearch), i.e., $G(W^{(k+1)}) - F(XW^{(k+1)}) \leq G(W^{(k)}) - F(XW^{(k)})$.

- If $G(W^{(k+1)}) - F(XW^{(k+1)}) = G(W^{(k)}) - F(XW^{(k)})$, then (DCA) terminates at the $k$th iteration and $W^{(k)}$ is a critical point of (P). Here, a critical point $\bar{W}$ of (P) is defined as a point satisfying $\partial G(\bar{W}) \cap X^\top \partial F(X\bar{W}) \neq \emptyset$, which is a necessary condition for optimality. If $G(W) - F(XW)$ is bounded below, then every limit point of the sequence $\{W^{(k)}\}_{k=0}^\infty$ is a critical point. We note that the optimal value being bounded below is an assumption that is trivially satisfied for all our upcoming PCA formulations.

- In general, under mild assumptions, the rate for (DCA) is sublinear. However, if certain growth conditions hold, then linear rates are possible, see for example [55, 43].

## 3 Principal Component Analysis

In this section, we briefly review the classical formulations of PCA before proposing novel DC dual pairs. In particular, we show that a PCA-like family with one of the functions in the primal being unitarily invariant, is kernelizable and out-of-sample applicable in the dual. Moreover, we look into (DCA) applied to these new formulations and provide a connection to simultaneous iteration.

### 3.1 Classical PCA

There are many formulations of PCA that are commonly used. Some of the most important ones are shown in Figure 1. Following the exposition of [54], we start from the fundamental low-rank approximation of the data matrix $X$ shown in (a). By the classical Eckart–Young–Mirsky theorem [26], the optimal solution is obtained by the sum of scaled outer products of left and right singular vectors, corresponding to the top $k$ singular values. This solution is unique if the spectrum of the covariance matrix is simple.

By parameterizing $A$ in (a) as its (full) SVD $U_A \Sigma_A V_A^\top$ where $U_A$ and $V_A$ are orthogonal matrices, one obtains the formulations (b) and (e) by taking $W = V_A$ and $H = U_A$ respectively. It should be noted that these problems do not admit unique solutions. In fact, there are infinitely many non-isolated minimizers since the problem is invariant to orthogonal transformations. This formulation is closely related to the Burer–Monteiro factorization for semidefinite programming [14, 47], though one key difference is that we impose orthogonality constraints on one of the factors.

To obtain formulation (c) from (b), it suffices to remark that (b) is a linear least-squares problem in the unconstrained factor $B$ and the unique solution is $B = XW$. The resulting formulation minimizes the reconstruction error of PCA as a linear autoencoder [58]. The formulation (f) is obtained from (e) in a completely analogous manner. By expanding the Frobenius norm in (c), the formulation (d) is obtained. This is the classical variance maximization objective of PCA, where it is common, though not necessary, to assume that each column has zero mean. The formulation (g) is similarly derived from (f) and now contains the kernel matrix $XX^\top$ (also known as the Gram matrix of the data). The relation between (d) and (g) lies at the heart of the success of many kernel methods.

Formulation (h) is derived by considering the low-rank approximation of the covariance matrix $X^\top X$, which is closely related to the formulations from the first column in Figure 1. To obtain (i), we impose the additional constraint that $A$ is positive semi-definite in (h) since it does not change the solution. A SVD of $A$ can now be written as $(U_A \Sigma_A^{1/2})(U_A \Sigma_A^{1/2})^\top$ which indeed yields the desired formulation. The last two formulations (j) and (k) follow immediately by expanding the square and using the definition of the Schatten 4-norm. It should be noted that four more formulations can be derived in an analogous manner by starting from the low-rank approximation of the kernel matrix $XX^\top$ instead.

### 3.2 DC dual formulations

This subsection is devoted to describing novel DC dual formulations of PCA. Our main result is the following.

**Theorem 3.1** (Fundamental PCA DC pairs). *Let $X \in \mathbb{R}^{N \times d}$ be the data matrix and $s \leq \mathrm{rank}(X)$. Then, the three DC dual pairs in Figure 2 hold. Moreover, (l) and (n) share the same minimizers, as do (o) and (q).*

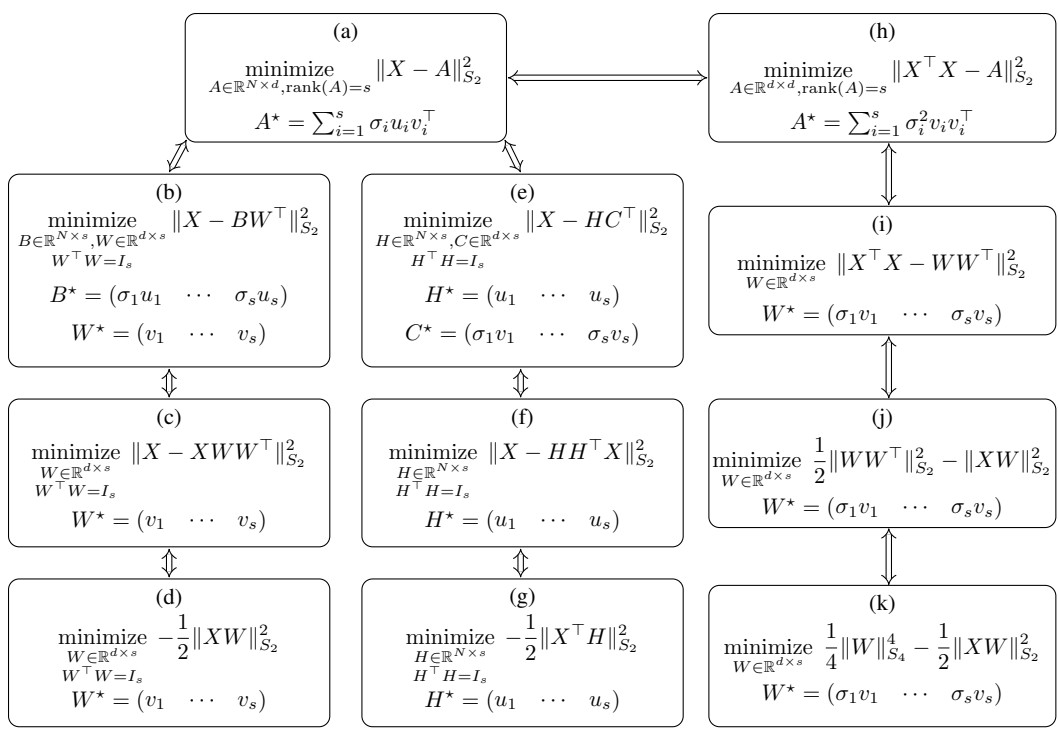

Figure 1: Classical PCA formulations for a data matrix $X \in \mathbb{R}^{N \times d}$ and $s \leq \mathrm{rank}(X)$ principal components. The starred variables denote (not necessarily unique) global minimizers of the associated formulation. A (full) SVD of $X$ is given by $U \, \mathrm{Diag}(\sigma) V^\top$. The arrows $\Leftrightarrow$ denote that the two formulations are 'equivalent', in the sense that one can easily obtain the minimizers of one problem by solving the other.

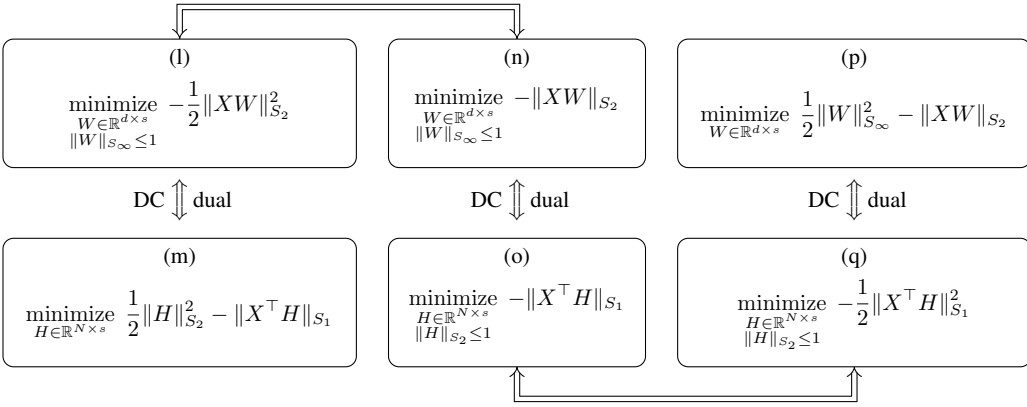

Figure 2: Three fundamental DC dual pairs for PCA.

To characterize the global minimizers of each problem from Figure 2, it suffices to combine the following known result with Proposition 2.3.

**Proposition 3.2.** *Let $X \in \mathbb{R}^{N \times d}$ be the data matrix and $s \leq \mathrm{rank}(X)$. Then, the optimal value of (l) is $-\frac{1}{2} \sum_{i=1}^{s} \sigma_i(X)^2$. Consider any eigenvalue decomposition $X^\top X = \tilde{W} \Lambda \tilde{W}^\top$, where $\Lambda = \overline{\mathrm{Diag}}(\lambda_1, \lambda_2, \ldots, \lambda_d)$ with $\lambda_1 \geq \lambda_2 \geq \cdots \geq \lambda_d$ and $\tilde{W}$ is a real orthogonal matrix. Then, the matrix $W \in \mathbb{R}^{d \times s}$ consisting of the first $s$ columns of $\tilde{W}$ achieves the optimum of (l).*

The DC dual pair (l)-(m) is well known for one component [9, 36, 5], and has recently been extended to multiple components in [74, Eq. (1) and (6)]. It should be noted that (l) is equivalent to (d) by

observing that the variance maximization formulation maximizes a (non-constant) convex function such that the constraint set can be relaxed to its convex hull, since the solutions necessarily lie on the boundary [61, Cor. 32.3.2]. Moreover, the constraint set is identified as the Stiefel manifold and its convex hull is the closed unit ball of the spectral norm [39, §3.4] which is exactly what appears in (l).

The two remaining DC dual pairs (n)-(o) and (p)-(q) are, to the best of our knowledge, novel. Their derivations rely on the simple fact that squaring/taking the square root and (positive) scaling does not change the solution sets of the minimization problems. Nevertheless, since the conjugates of the resulting transformed problems are quite different, so are their DC duals. In particular, the DC dual pair (n)-(o) is arguably more fundamental due to its symmetry. Each formulation captures a specific trade-off between Schatten $p$-norms.

Another interesting and novel dual problem follows from starting from (k) in Figure 1.

**Proposition 3.3.** *Let $X \in \mathbb{R}^{N \times d}$ be the data matrix and $s \leq \operatorname{rank}(X)$. Then, the following DC dual pair holds.*

$$\operatorname*{minimize}_{W \in \mathbb{R}^{d \times s}} \frac{1}{4}\|W\|_{S_4}^4 - \frac{1}{2}\|XW\|_{S_2}^2 \quad \overset{\text{DC dual}}{\Longleftrightarrow} \quad \operatorname*{minimize}_{H \in \mathbb{R}^{N \times s}} \frac{1}{2}\|H\|_{S_2}^2 - \frac{3}{4}\|X^\top H\|_{S_{4/3}}^{4/3}.$$

**Remark 3.4.** By replacing $X$ with $X^\top$ and vice versa (cf. (d) and (g)), four more analogous DC dual pairs can be formulated as the ones from Theorem 3.1 and Proposition 3.3.

In the Hilbert space setting, formulations involving the kernel matrix $K := XX^\top$ are needed, as they enable computations via the kernel trick. This is the case when $G$ in the primal is unitarily invariant as we show in the following proposition.

**Proposition 3.5** (Kernelizability for unitarily invariant $G$). *Let $X \in \mathbb{R}^{N \times d}$ be the data matrix and $s \leq \operatorname{rank}(X)$. Consider the primal problem (P) and suppose $G = g \circ \sigma$ is unitarily invariant. Then, the corresponding DC dual (D) is kernelizable. In particular, (D) can be written as*

$$\operatorname*{minimize}_{H \in \mathbb{R}^{N \times s}} F^*(H) - g^* \left( \sqrt{\lambda(H^\top K H)} \right)$$

*where $\sqrt{\cdot}$ is taken elementwise and $\lambda(\cdot)$ denotes the vector of eigenvalues of its argument (in any order).*

**Remark 3.6.** The decomposition $G = g \circ \sigma$ always exists for unitarily invariant functions, where moreover $g$ is absolutely symmetric, see Fact A.4. This fact also ensures that $g^*(\cdot)$ does not depend on the order of its arguments.

**Remark 3.7.** Note that while the objective function contains an eigenvalue decomposition, it is of an $s \times s$ matrix, which is small in practice, as typically only a few principal components are required. In particular, when $s = 1$, we have that $\lambda(H^\top K H) = H^\top K H \in \mathbb{R}$.

The previous result allows us to characterize a wide variety of formulations that are kernelizable in the DC dual, making the computations tractable. In particular, all the previously described formulations are kernelizable, since they are based on Schatten $p$-norms, which are unitarily invariant.

Another important requirement in practical applications is handling new, unseen data. The following result shows how one can perform feature extraction of a new sample by projecting onto the associated primal (lower-dimensional) subspace using only operations that are compatible with the kernel trick, thereby yielding out-of-sample applicability.

**Theorem 3.8** (Out-of-sample applicability for unitarily invariant $G$). *Let $X \in \mathbb{R}^{N \times d}$ be the data matrix, $K = XX^\top \in \mathbb{R}^{N \times N}$ the kernel matrix, $s \leq \operatorname{rank}(X)$ the number of principal components and $\tilde{x} \in \mathbb{R}^d$ a (new) data point. Consider the primal problem (P) and suppose $G = g \circ \sigma$ is unitarily invariant. Suppose moreover that $H^\star \in \mathbb{R}^{N \times s}$ is a minimizer of the corresponding DC dual (D) and $H^{\star\top} K H^\star$ is nonsingular. Denote the eigenvalue decomposition of $H^{\star\top} K H^\star \in \mathbb{R}^{s \times s}$ as $V^\top \overline{\operatorname{Diag}}(\lambda) V$. Then, there exists a $W^\star$ which is a solution of (P) such that*

$$W^{\star\top} \tilde{x} = V \overline{\operatorname{Diag}}(\mu \odot \lambda^{-1/2}) V^\top H^{\star\top} X \tilde{x}$$

*where $\mu \in \partial g^*(\sqrt{\lambda})$, $\odot$ denotes the elementwise product, and both $\sqrt{\lambda}$ and $\lambda^{-1/2}$ are to be understood elementwise.*

**Remark 3.9.** In a practical implementation, after finding an optimal solution $H^\star$ based on the training data, the matrix $V \overline{\operatorname{Diag}}(\mu \odot \lambda^{-1/2}) V^\top H^{\star\top} \in \mathbb{R}^{s \times N}$ only needs to be calculated once.

### 3.3 DC algorithms

We now derive the algorithm (DCA) for multiple formulations from Subsection 3.2. To this end, a subgradient of a unitarily invariant function $G = g \circ \sigma$ is needed, which we can calculate from the following known result.

**Proposition 3.10** ([46, Cor. 2.5]). *Let $X \in \mathbb{R}^{m \times n}$ and let $g : \mathbb{R}^{\min(m,n)} \to \overline{\mathbb{R}}$ be a proper and absolutely symmetric function. Suppose moreover that $\sigma(X) \subseteq \mathrm{dom}\, g$. Then,*

$$\partial(g \circ \sigma)(X) = \{U \operatorname{Diag}(\mu) V^\top \mid \mu \in \partial g(\sigma(X)),\ U \operatorname{Diag}(\sigma(X)) V^\top = \mathrm{SVD}(X)\}.$$

By now applying (DCA) to the formulations (d) and (g) with the constraint set relaxed to its convex hull, one obtains Algorithms 1 and 2. Here $\overline{\mathbf{SVD}}$ denotes a compact SVD, i.e., for $[U, \Sigma, V^\top] = \overline{\mathbf{SVD}}(X)$ with $X \in \mathbb{R}^{m \times n}$, we have $U \in \mathbb{R}^{m \times r}$ and $V^\top \in \mathbb{R}^{r \times n}$ where $r$ denotes the rank of $X$, and $U^\top U = I_r = V^\top V$. The derivation of these algorithms, as well as the reason why a compact SVD can be used here, is detailed in Appendix C.

| **Algorithm 1** (DCA) for (d) | **Algorithm 2** (DCA) for (g) |
|---|---|
| **Require:** Matrix $X \in \mathbb{R}^{N \times d}$ | **Require:** Matrix $K = XX^\top \in \mathbb{R}^{N \times N}$ |
| 1: Initialize $W^{(0)} \in \mathbb{R}^{d \times s}$ s.t. $\|W^{(0)}\|_{S_\infty} = 1$ | 1: Initialize $H^{(0)} \in \mathbb{R}^{N \times s}$ s.t. $\|H^{(0)}\|_{S_\infty} = 1$ |
| 2: Initialize $k \leftarrow 0$ | 2: Initialize $k \leftarrow 0$ |
| 3: **repeat** | 3: **repeat** |
| 4: $\quad [U^{(k)}, \Sigma^{(k)}, V^{(k)^\top}] \leftarrow \overline{\mathbf{SVD}}(X^\top X W^{(k)})$ | 4: $\quad [U^{(k)}, \Sigma^{(k)}, V^{(k)^\top}] \leftarrow \overline{\mathbf{SVD}}(K H^{(k)})$ |
| 5: $\quad W^{(k+1)} \leftarrow U^{(k)} V^{(k)^\top}$ | 5: $\quad H^{(k+1)} \leftarrow U^{(k)} V^{(k)^\top}$ |
| 6: $\quad k \leftarrow k + 1$ | 6: $\quad k \leftarrow k + 1$ |
| 7: **until** convergence | 7: **until** convergence |

Observe that when $s = 1$, the SVD is applied to a vector, resulting in its normalization, and the iterations correspond to the power method. This fact was also observed in [71, Prop. 4] after removing the sparsity constraint. In the next result, we show that these algorithms are connected to simultaneous iteration [77, Algorithm 28.3] which is strongly related to the QR algorithm [77, Theorem 28.3].

**Theorem 3.11.** *Algorithms 1 and 2 correspond to simultaneous iteration applied to $X^\top X$ and $XX^\top$ respectively, up to orthogonal transformation.*

Although the previous result states that (DCA) identifies the same subspace as the simultaneous iteration method, whose iterates span the leading eigenvector basis, (DCA) does not necessarily converge to the eigenvectors themselves, as the subspace admits infinitely many (orthonormal) bases. For the purpose of obtaining a lower-dimensional representation with meaningful features, the projection onto this subspace is often sufficient. If the principal vectors are specifically required, they can be recovered by performing an eigenvalue decomposition of a smaller $s \times s$ matrix.

An immediate corollary of this result and the known convergence result for simultaneous iteration [77, Theorem 28.4] is that Algorithms 1 and 2 inherit the linear rate as well as the convergence to global optimality with probability one, starting from random initialization. Similarly, the proximal gradient method that we use in Section 5 has a well-known interpretation of (DCA) applied to $(G + (1/2)\|\cdot\|^2) - (F + (1/2)\|\cdot\|^2)$, where $G$ denotes the indicator of a convex set. Therefore, these can be connected to simultaneous iteration on an identity-shifted covariance/kernel matrix (i.e., it naturally incorporates regularization), such that again convergence to global optima is guaranteed.

Applying (DCA) to the problems (m) through (q) as well as their transposed variants (cf. Remark 3.4) is deferred to Appendix C. These algorithms are equivalent up to normalization and simple operations, as is to be expected due to the elementary problem transformation. A more interesting algorithm follows from considering (DCA) for the problems from Proposition 3.3. These are presented in Algorithms 3 and 4. The iteration is similar but now also uses information from the singular values through some nonlinear preconditioning. Here $\overline{\mathbf{EIG}}$ denotes an eigenvalue decomposition of its argument but only keeping the columns corresponding to nonzero eigenvalues. Note that we are able to obtain a kernelizable formulation in the dual. The derivation is deferred to Appendix C.

| **Algorithm 3** (DCA) for Proposition 3.3 (primal) | **Algorithm 4** (DCA) for Proposition 3.3 (dual) |
|---|---|
| **Require:** Matrix $X \in \mathbb{R}^{N \times d}$ | **Require:** Matrix $K = XX^\top \in \mathbb{R}^{N \times N}$ |
| 1: Initialize $W^{(0)} \in \mathbb{R}^{d \times s}$ | 1: Initialize $H^{(0)} \in \mathbb{R}^{N \times s}$ |
| 2: Initialize $k \leftarrow 0$ | 2: Initialize $k \leftarrow 0$ |
| 3: **repeat** | 3: **repeat** |
| 4: $\quad [U^{(k)}, \Sigma^{(k)}, V^{(k)^\top}] \leftarrow \overline{\mathbf{SVD}}(X^\top X W^{(k)})$ | 4: $\quad [V^{(k)}, \Lambda^{(k)}] \leftarrow \overline{\mathbf{EIG}}(H^{(k)^\top} K H^{(k)})$ |
| 5: $\quad W^{(k+1)} \leftarrow U^{(k)}(\Sigma^{(k)})^{1/3} V^{(k)^\top}$ | 5: $\quad H^{(k+1)} \leftarrow K H^{(k)} V^{(k)} (\Lambda^{(k)})^{-1/3} V^{(k)^\top}$ |
| 6: $\quad k \leftarrow k + 1$ | 6: $\quad k \leftarrow k + 1$ |
| 7: **until** convergence | 7: **until** convergence |

## 4 Robust PCA

In this section, we start from formulation (c), the reconstruction error interpretation of PCA as a linear autoencoder, and note that it can be equivalently written as

$$\underset{\substack{W \in \mathbb{R}^{d \times s} \\ W^\top W = I_s}}{\text{minimize}} \sum_{i=1}^{N} \|X_{i,:} - WW^\top X_{i,:}\|^2$$

where $\|\cdot\|$ denotes the Euclidean norm. It is well known that measuring the error through the squared loss leads to non-robust models. A popular approach to obtain a robust version is to remove the square, thereby considering an $l_1$ penalty on the reconstruction errors instead. This is known in the robust subspace recovery literature as the least absolute deviation formulation [45]. We have the following DC dual pair, which was considered in [9] for only one component.

**Proposition 4.1.** *Let $X \in \mathbb{R}^{N \times d}$ be the data matrix and $s \leq \text{rank}(X)$. Then, the following DC dual pair holds:*

$$\underset{\substack{W \in \mathbb{R}^{d \times s} \\ \|W\|_{S_\infty} \leq 1}}{\text{minimize}} \sum_{i=1}^{N} \|X_{i,:} - WW^\top X_{i,:}\| \overset{\text{DC dual}}{\Longleftrightarrow} \underset{H \in \mathbb{R}^{N \times s}}{\text{minimize}} \left( \sum_{i=1}^{N} \|X_{i,:}\| \sqrt{1 + \|H_{i,:}\|^2} \right) - \|X^\top H\|_{S_1}.$$

The dual formulation is kernelizable, as it can be written as

$$\underset{H \in \mathbb{R}^{N \times s}}{\text{minimize}} \left( \sum_{i=1}^{N} \sqrt{(XX^\top)_{i,i}(1 + \|H_{i,:}\|^2)} \right) - \text{tr}\left( \sqrt{H^\top XX^\top H} \right).$$

Moreover, since the associated $G$ is the indicator function of the spectral unit norm ball, which is unitarily invariant, it satisfies the assumptions of Theorem 3.8 and therefore the problem is out-of-sample applicable. By applying (DCA) to both formulations from Proposition 4.1, we obtain Algorithms 5 and 6. In Appendix D, we show how the primal algorithm is related to an iteratively reweighted least squares scheme.

| **Algorithm 5** (DCA) for Proposition 4.1 (primal) | **Algorithm 6** (DCA) for Proposition 4.1 (dual) |
|---|---|
| **Require:** Matrix $X \in \mathbb{R}^{N \times d}$ | **Require:** Matrix $K = XX^\top \in \mathbb{R}^{N \times N}$ |
| 1: Initialize $W^{(0)} \in \mathbb{R}^{d \times s}$ s.t. $\|W^{(0)}\|_{S_\infty} = 1$ | 1: Initialize $H^{(0)} \in \mathbb{R}^{N \times s}$ |
| 2: Initialize $k \leftarrow 0$ | 2: Initialize $k \leftarrow 0$ |
| 3: **repeat** | 3: **repeat** |
| 4: $\quad$ **for** $i = 1$ **to** $N$ **do** | 4: $\quad [V^{(k)}, \Lambda^{(k)}] \leftarrow \overline{\mathbf{EIG}}(H^{(k)^\top} K H^{(k)})$ |
| 5: $\quad\quad Y_{i,:}^{(k)} \leftarrow \dfrac{(XW^{(k)})_{i,:}}{\sqrt{\|X_{i,:}\|^2 - \|(XW^{(k)})_{i,:}\|^2}}$ | 5: $\quad Y^{(k)} \leftarrow K H^{(k)} V^{(k)} (\Lambda^{(k)})^{-1/2} V^{(k)^\top}$ |
| 6: $\quad$ **end for** | 6: $\quad$ **for** $i = 1$ **to** $N$ **do** |
| 7: $\quad [U^{(k)}, \Sigma^{(k)}, V^{(k)^\top}] \leftarrow \overline{\mathbf{SVD}}(X^\top Y^{(k)})$ | 7: $\quad\quad H_{i,:}^{(k)} \leftarrow \dfrac{(Y^{(k)})_{i,:}}{\sqrt{K_{i,i} - \|(Y^{(k)})_{i,:}\|^2}}$ |
| 8: $\quad W^{(k+1)} \leftarrow U^{(k)} V^{(k)^\top}$ | 8: $\quad$ **end for** |
| 9: $\quad k \leftarrow k + 1$ | 9: $\quad k \leftarrow k + 1$ |
| 10: **until** convergence | 10: **until** convergence |

Table 1: Timing results for various methods applied to PCA formulations. The problem setting $(N, d, s, \epsilon)$ denotes a data matrix $X \in \mathbb{R}^{N \times d}$ with entries sampled from a standard normal distribution. $s$ denotes the computed number of principal components and $\varepsilon$ the stopping criterion tolerance. All timings are in milliseconds, and timings longer than 5 seconds are not displayed.

| Method | $(4000, 2000, 20, 10^{-3})$ | $(2000, 4000, 20, 10^{-3})$ | $(4500, 4500, 20, 10^{-3})$ |
|---|---|---|---|
| ZeroFPR (l) | $2828.1 \pm 376.5$ | $1949.3 \pm\ \ 97.9$ | $8719.1$ |
| ZeroFPR (n) | $210.4 \pm\ \ 54.9$ | $196.3 \pm\ \ 49.1$ | $441.0 \pm\ \ 86.3$ |
| ZeroFPR (o) | $392.6 \pm 100.7$ | $463.0 \pm 121.5$ | $1130.7 \pm 143.3$ |
| ZeroFPR (q) | $2341.7 \pm 310.4$ | $2021.0 \pm 205.4$ | / |
| PG (l) | / | / | / |
| PG (n) | $207.5 \pm\ \ 14.0$ | $139.1 \pm\ \ 42.4$ | $309.1 \pm\ \ 26.1$ |
| PG (o) | $203.6 \pm\ \ 57.4$ | $216.4 \pm\ \ 29.1$ | $689.7 \pm\ \ 53.4$ |
| PG (q) | / | / | / |
| Algorithm 1 | $640.1 \pm 154.5$ | $2630.9 \pm\ \ 33.0$ | $3696.4 \pm 115.2$ |
| Algorithm 2 | $2218.5 \pm\ \ 35.2$ | $635.3 \pm\ \ 85.1$ | $3294.2 \pm 359.8$ |
| Algorithm 3 | $3848.8 \pm 393.3$ | / | / |
| Algorithm 4 | / | $1868.2 \pm 182.6$ | / |
| SVDS | $808.0 \pm\ \ 37.6$ | $808.2 \pm\ \ 29.6$ | $2985.8 \pm 128.3$ |
| KrylovKit | $1130.4 \pm\ \ 21.6$ | $940.1 \pm\ \ 13.6$ | $3844.6 \pm 253.2$ |

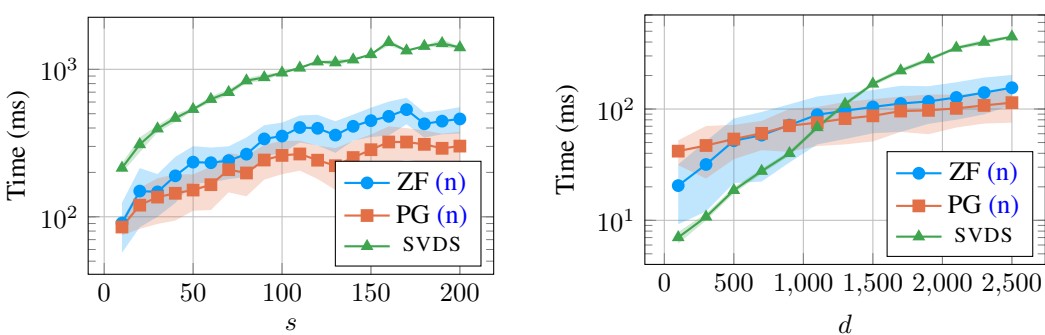

Figure 3: (left) Timings for the problem setting $(2000, 2000, s, 10^{-3})$ as defined in Table 1, with varying $s$ and one standard deviation error bars. (right) Timings for the problem setting $(2000, d, 30, 10^{-3})$ as defined in Table 1, with varying $d$ and one standard deviation error bars. In both figures, ZF is shorthand for ZeroFPR.

## 5 Experiments

We compare several simple gradient-based methods on various formulations from Theorem 3.1 and Proposition 3.3. All experiments are implemented in Julia 1.11.1 on a machine with an AMD Ryzen 7 Pro 5850U processor and 32 GB RAM. The timings are taken using BenchmarkTools.jl [17]. Our results are presented in Table 1. The code for reproducing all experiments is publicly available[2].

For the constrained problems, we apply both the proximal gradient (PG) method and ZeroFPR [70] using the implementation from [65]. Regarding the convergence properties of these algorithms, the standard assumptions to minimize objectives of the form $f(x) + g(x)$ using these methods require $f$ to be $L$-smooth. In general, the (global) rates of PG and ZeroFPR to stationary points are both sublinear [70]. We note that, strictly speaking, for some formulations our $f$ is nonsmooth. However, this does not pose a problem since the objectives are concave. Therefore, the Euclidean descent lemma [8, Lemma 5.7] holds with $L = 0$, and both PG and ZeroFPR enjoy the same theoretical guarantees as described above. In fact, we may choose arbitrarily large stepsizes, despite nonsmoothness. As a baseline, we compare our formulations against classical eigensolvers. Specifically, we used the classical ARPACK implementation of SVDS, which is based on the implicitly restarted Arnoldi

[2]https://github.com/JanQ/pca-duality

method [44]. Moreover, we also compare with KrylovKit.jl, a state-of-the-art package containing efficient and stable implementations of various Krylov subspace methods [33].

In the setting of this experiment, we conclude that the performance of PG and ZeroFPR varies depending on the formulation that is considered, thus motivating the theoretical framework presented in Section 3. In particular, state-of-the-art performance is achieved when using the newly proposed formulations (n)-(o). Moreover, we observe that depending on the shape of the data matrix, either the primal or the dual formulation is preferred. This is to be expected since the number of decision variables varies accordingly. We also remark that for classical variance-based formulations, these first-order methods do not perform well.

We also investigate the effect of the number of principal components and the sizes of the matrices involved for the best-performing methods in Figure 3. It can be seen that the first-order methods consistently outperform the classical methods for varying number of principal components, provided that the dimensions of the involved matrices are sufficiently large.

Additional experiments showcasing the performance of these first-order methods on various formulations as well as toy experiments for our robust kernel PCA can be found in Appendix F.

## 6 Limitations

We showed that the dual is kernelizable if one of the functions in the primal is unitarily invariant. While this may appear restrictive, many interesting functions such as the magnitude of the determinant, Ky Fan norms and Schatten $p$-norms all fall under this category, thereby leading to a whole new family of kernelizable formulations, derived in a principled way. Another issue that many kernel-based methods have is scalability, since the size of the kernel matrix scales as $N^2$. Nevertheless, many mitigations have been proposed to deal with this problem, such as Nyström approximations [80, 30] or random Fourier features [60], which are also applicable to our framework.

Another limitation is that (DCA) does not yield the eigenvectors but only an (orthonormal) basis for the eigenspace. However, this is not really an issue in practice since only a small number of principal components are typically required and the eigenbasis can be recovered in a postprocessing step.

## 7 Conclusion and future outlook

In this paper we revisited PCA under the light of difference-of-convex duality. First, we proposed several novel DC pairs that yield more insight into the inner workings of PCA. Moreover, we showed that when one of the terms in the primal is unitarily invariant, the corresponding dual is not only kernelizable but also supports out-of-sample extensions which is essential for modern machine learning applications. Further, we showed that applying DCA to the variance maximization is related to simultaneous iteration, revealing a novel connection between optimization and numerical linear algebra. In addition, we derived a new kernelizable DC dual for robust kernel principal component analysis and an associated algorithm, which in turn is connected to iteratively reweighted least squares. Lastly, our experimental results showed that for the correct formulation, simple first-order optimization methods can outperform state-of-the-art solvers, depending on the required accuracy.

Several interesting research directions for future work remain open. One promising avenue is the exploration of other unitarily invariant objectives in the primal by incorporating domain knowledge or structural priors of the problem at hand. An alternative research direction is to consider deep variants by stacking (kernel) PCA layers [75] where each layer can be described with different formulations. In addition, a further line of research could involve developing specialized algorithms for our new DC dual pairs, as currently most of the algorithms are derived for (l) in Figure 2.

## Acknowledgments and Disclosure of Funding

This work was supported by the Research Foundation Flanders (FWO) PhD grant 11A8T26N and research projects G081222N, G033822N, G0A0920N; Research Council KUL grant C14/24/103, iBOF/23/064; Flemish Government AI Research Program.

The authors thank Alexander Bodard for assistance with the experiments.

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

# A Facts from convex analysis

We include a summary of relevant facts from convex analysis in this Appendix for completeness.

**Fact A.1** (Equality case of Fenchel–Young inequality, [7, Theorem 16.29]). *Let $\mathcal{H}$ be a Hilbert space with inner product $\langle \cdot, \cdot \rangle_{\mathcal{H}}$. Let $f : \mathcal{H} \to \overline{\mathbb{R}}$ be a convex, closed and proper function. Let $x, u \in \mathcal{H}$, then the following are equivalent:*

$$u \in \partial f(x) \iff f(x) + f^*(u) = \langle x, u \rangle_{\mathcal{H}} \iff x \in \partial f^*(u).$$

**Fact A.2** (Conjugate separable sum, [8, Theorem 4.12]). *Let $f : \mathbb{R}^{n_1} \times \mathbb{R}^{n_2} \times \cdots \times \mathbb{R}^{n_N} \to \overline{\mathbb{R}}$ be given by $f(x_1, x_2, \ldots, x_N) = \sum_{i=1}^{N} f_i(x_i)$ where each $f_i : \mathbb{R}^{n_i} \to \overline{\mathbb{R}}$ is proper. Then $f^*(y_1, y_2, \ldots, y_N) = \sum_{i=1}^{N} f_i^*(y_i)$.*

**Fact A.3** (Conjugates). *The conjugate pairs in Table 2 hold. Recall that the indicator function $\delta_C$ of a set $C$ is given by*

$$\delta_C(x) = \begin{cases} 0 & x \in C, \\ +\infty & \text{otherwise.} \end{cases}$$

Table 2: Conjugate functions [8, Appendix B]. $\| \cdot \|_*$ denotes the dual norm of $\| \cdot \|$ and $\| \cdot \|_p$ denotes the $l_p$-norm.

| $f$ | $\mathrm{dom}(f)$ | $f^*$ | Assumption | |
|---|---|---|---|---|
| $\frac{1}{2}\|x\|^2$ | $\mathbb{R}^n$ | $\frac{1}{2}\|y\|_*^2$ | - | (1) |
| $\delta_{B_{\|\cdot\|}}(x)$ | $B_{\|\cdot\|}$ | $\|y\|_*$ | - | (2) |
| $\frac{1}{p}\|x\|_p^p$ | $\mathbb{R}^n$ | $\frac{1}{q}\|x\|_*^q$ | $p > 1, \frac{1}{p} + \frac{1}{q} = 1$ | (3) |
| $-\sqrt{\alpha^2 - \|x\|^2}$ | $B_{\alpha^{-1}\|\cdot\|}$ | $\alpha\sqrt{\|y\|_*^2 + 1}$ | $\alpha > 0$ | (4) |

**Fact A.4** (Equivalence unitarily invariant and absolutely symmetric, [46, Prop. 2.2]). *A function $F : \mathbb{R}^{m \times n} \to \overline{\mathbb{R}}$ is unitarily invariant if and only if it can be written as $f \circ \sigma$ where $f : \mathbb{R}^q \to \overline{\mathbb{R}}$ is absolutely symmetric and $\sigma : \mathbb{R}^{m \times n} \to \mathbb{R}^q$ denotes the singular value map with $q = \min(m, n)$, i.e., $\sigma(X)$ is the vector with components $\sigma_1(X) \geq \sigma_2(X) \geq \ldots \geq \sigma_q(X) \geq 0$, the singular values of $X$.*

**Fact A.5** (Spectral conjugate, [46, Theorem 2.4]). *Let $g : \mathbb{R}^q \to \overline{\mathbb{R}}$ be absolutely symmetric. Then*

$$(g \circ \sigma)^* = g^* \circ \sigma$$

*where $\sigma$ is defined as in Fact A.4.*

**Corollary A.6** (Conjugates spectral functions). *The conjugate pairs in Table 3 hold.*

Table 3: Conjugates of spectral functions.

| $f$ | $\mathrm{dom}(f)$ | $f^*$ | Assumption | |
|---|---|---|---|---|
| $\delta_{B_{\|\cdot\|_{S_\infty}}}(X)$ | $B_{\|\cdot\|_{S_\infty}}$ | $\|Y\|_{S_1}$ | - | (5) |
| $\|X\|_{S_p}$ | $\mathbb{R}^{m \times n}$ | $\delta_{B_{\|\cdot\|_{S_q}}}(Y)$ | $p > 1, \frac{1}{p} + \frac{1}{q} = 1$ | (6) |
| $\frac{1}{2}\|X\|_{S_\infty}^2$ | $\mathbb{R}^{m \times n}$ | $\frac{1}{2}\|Y\|_{S_1}^2$ | - | (7) |
| $\frac{1}{p}\|X\|_{S_p}^p$ | $\mathbb{R}^{m \times n}$ | $\frac{1}{q}\|Y\|_{S_q}^q$ | $p > 1, \frac{1}{p} + \frac{1}{q} = 1$ | (8) |

**Fact A.7** (Subdifferential composition with norm, [7, Example 16.73]). *Let $h : \mathbb{R} \to \mathbb{R}$ be convex and even, and let $f = h \circ \| \cdot \|$. Then, $\partial h(0) = [-\rho, \rho]$ for some $\rho \in \mathbb{R}_+$ and*

$$\partial f(x) = \begin{cases} \left\{ \frac{\alpha}{\|x\|} x \mid \alpha \in \partial h(\|x\|) \right\} & \text{if } x \neq 0, \\ B_{\rho^{-1}\|\cdot\|} & \text{if } x = 0. \end{cases}$$

# B Missing proofs

## B.1 Proof of Proposition 2.3

We recall the proof from [73, Sec. 3.1] for completeness.

*Proof.* Let $\mathcal{H}_1 = \mathbb{R}^{d \times s}$, $\mathcal{H}_2 = \mathbb{R}^{N \times s}$ and denote $\langle \cdot, \cdot \rangle_{\mathcal{H}_1}, \langle \cdot, \cdot \rangle_{\mathcal{H}_2}$ for the corresponding inner products. Since $F$ is convex, closed and proper, we have that $F = F^{**}$ [7, Cor. 13.38]. Combining this fact with the definition of the convex conjugate yields

$$
\begin{aligned}
\inf_{W \in \mathcal{H}_1} G(W) - F(XW) &= \inf_{W \in \mathcal{H}_1} G(W) - F^{**}(XW) \\
&= \inf_{W \in \mathcal{H}_1} G(W) - \sup_{H \in \mathcal{H}_2} \{ \langle H, XW \rangle_{\mathcal{H}_2} - F^*(H) \} \\
&= \inf_{W \in \mathcal{H}_1} \inf_{H \in \mathcal{H}_2} G(W) + F^*(H) - \langle H, XW \rangle_{\mathcal{H}_2} \\
&= \inf_{H \in \mathcal{H}_2} F^*(H) - \sup_{W \in \mathcal{H}_1} \{ \langle X^\top H, W \rangle_{\mathcal{H}_1} - G(W) \} \\
&= \inf_{H \in \mathcal{H}_2} F^*(H) - G^*(X^\top H)
\end{aligned}
$$

which shows the strong duality of the two problems. Moreover, let $W^\star$ be a solution of (P), i.e.,

$$
G(W^\star) - F(XW^\star) = \min_{W \in \mathcal{H}_1} G(W) - F(XW) = \inf_{H \in \mathcal{H}_2} F^*(H) - G^*(X^\top H).
$$

Thus we have

$$
G(W^\star) - F(XW^\star) \leq F^*(\tilde{H}) - G^*(X^\top \tilde{H}) \tag{9}
$$

for all $\tilde{H} \in \mathcal{H}_2$. Let $H^\star \in \partial F(XW^\star)$, then by Fact A.1

$$
F(XW^\star) + F^*(H^\star) = \langle XW^\star, H^\star \rangle_{\mathcal{H}_2} = \langle W^\star, X^\top H^\star \rangle_{\mathcal{H}_1}
$$

which we substitute in (9) with $\tilde{H} = H^\star$ to obtain

$$
G(W^\star) - \langle W^\star, X^\top H^\star \rangle_{\mathcal{H}_1} \leq -G^*(X^\top H^\star) \iff G(W^\star) + G^*(X^\top H^\star) \leq \langle W^\star, X^\top H^\star \rangle_{\mathcal{H}_1}.
$$

The Fenchel–Young inequality [7, Prop. 13.15] yields

$$
G(W^\star) + G^*(X^\top H^\star) \geq \langle W^\star, X^\top H^\star \rangle_{\mathcal{H}_1}
$$

such that we must have equality throughout and therefore

$$
G(W^\star) - F(XW^\star) = F^*(H^\star) - G^*(X^\top H^\star)
$$

which shows that $H^\star$ solves (D) since strong duality holds. The proof that any $W^\star \in \partial G^*(X^\top H^\star)$ is a solution of (P) whenever $H^\star$ is a solution of (D) is completely analogous. $\square$

## B.2 Proof of Proposition 2.5

*Proof.* The iterates for (DCA) applied to (P) and (D) are respectively

$$
\begin{aligned}
H^{(k)} &\in \partial F(XW^{(k)}); & W^{(k+1)} &\in \partial G^*(X^\top H^{(k)}), \\
\tilde{W}^{(k)} &\in \partial G^*(X^\top \tilde{H}^{(k)}); & \tilde{H}^{(k+1)} &\in \partial F^{**}(X\tilde{W}^{(k)}) = \partial F(X\tilde{W}^{(k)})
\end{aligned}
$$

where the latter equality follows from the fact that $F$ is convex, closed and proper and therefore $F = F^{**}$ [7, Cor. 13.38]. We now proceed by induction on $k$. The base case follows immediately by our choice of $\tilde{H}^{(0)} \in \partial F(XW^{(0)})$. Suppose that the statement holds for $k \geq 0$, we show that it also holds for $k + 1$. Indeed, since $W^{(k+1)} \in \partial G^*(X^\top \tilde{H}^{(k)})$ by the induction hypothesis, we may take $\tilde{W}^{(k)} = W^{(k+1)}$ and

$$
\tilde{H}^{(k+1)} = H^{(k+1)} \in \partial F(XW^{(k+1)})
$$

which implies

$$
W^{(k+2)} \in \partial G^*(X^\top H^{(k+1)}) = \partial G^*(X^\top \tilde{H}^{(k+1)}),
$$

as desired. $\square$

### B.3 Proof of Theorem 3.1

*Proof.* The problem (l) can be written in the form (P) via $G = \delta_{B_{\|\cdot\|_{S_\infty}}}$ and $F = \frac{1}{2}\|\cdot\|_{S_2}^2$. The DC dual follows from Proposition 2.3, (5) and (8).

The problem (n) is equivalent to (l) since multiplying by two and taking the square root does not change the set of minimizers. The problem (n) can be written in the form (P) via $G = \delta_{B_{\|\cdot\|_{S_\infty}}}$ and $F = \|\cdot\|_{S_2}$. The DC dual follows from Proposition 2.3, (5) and (6).

The problem (o) is equivalent to (q) since squaring the norm and dividing by two does not change the set of minimizers. The problem (p) can be written in the form (P) via $G = \frac{1}{2}\|\cdot\|_{S_\infty}^2$ and $F = \|\cdot\|_{S_2}$. The DC dual follows from Proposition 2.3, (7) and (6). $\square$

### B.4 Proof of Proposition 3.2

*Proof.* Since the solution of (l) lies at the boundary by the discussion after Proposition 3.2, we may consider the equivalent problem

$$\underset{\substack{W\in\mathbb{R}^{d\times s}\\ W^\top W=I}}{\text{minimize}}\ F(W) := -\frac{1}{2}\|XW\|_{S_2}^2 = -\frac{1}{2}\operatorname{tr}(W^\top X^\top X W). \tag{10}$$

Writing an eigenvalue decomposition $X^\top X = \tilde{W}\overline{\operatorname{Diag}}(\Lambda)\tilde{W}^\top$ with $\tilde{W}$ real orthogonal, and assuming that the (necessarily nonnegative real) eigenvalues are in descending order, we find that $\overline{\operatorname{Diag}}(\Lambda)$ contains the entries $\sigma_1(X)^2 \geq \cdots \geq \sigma_d(X)^2 \geq 0$ on the diagonal by [37, Theorem 2.6.3(b)]. Letting $W^\star$ be the matrix consisting of the first $s$ columns of $\tilde{W}$, and using the orthonormality of the columns of $\tilde{W}$, we obtain that

$$F(W^\star) = -\frac{1}{2}\operatorname{tr}(W^{\star\top}\tilde{W}\overline{\operatorname{Diag}}(\sigma_1(X)^2,\ldots,\sigma_d(X)^2)\tilde{W}^\top W^\star) = -\frac{1}{2}\sum_{i=1}^s \sigma_i(X)^2$$

so the minimal value of (10) is less than or equal to $F(W^\star)$. On the other hand, from [37, Theorem 4.3.45], we find that $F(W) \geq -\frac{1}{2}\sum_{i=1}^s \sigma_i(X)^2$ such that $W^\star$ is a solution of (10). $\square$

### B.5 Proof of Proposition 3.3

*Proof.* The problem (k) can be written in the form (P) via $G = \frac{1}{4}\|\cdot\|_{S_4}^4$ and $F = \frac{1}{2}\|\cdot\|_{S_2}^2$. The DC dual follows from Proposition 2.3 and (8). $\square$

### B.6 Proof of Proposition 3.5

*Proof.* Since $G$ is absolutely invariant, it can be written as $g\circ\sigma$ by Fact A.4. Then, by using Proposition 2.3 and Fact A.5, we find that (D) becomes

$$\underset{H\in\mathbb{R}^{N\times s}}{\text{minimize}}\ F^*(H) - (g^*\circ\sigma)(X^\top H).$$

But the singular value vector $\sigma(X^\top H)$ is equal to the elementwise square root of the nonincreasing eigenvalue vector $\lambda(H^\top KH)$, i.e., $\sigma(X^\top H) = \sqrt{\lambda(H^\top KH)}$. Therefore, the DC dual (D) takes the form

$$\underset{H\in\mathbb{R}^{N\times s}}{\text{minimize}}\ F^*(H) - g^*\left(\sqrt{\lambda(H^\top KH)}\right)$$

where $\lambda(X)$ denotes the vector with the eigenvalues of $X$ (necessarily square) in any order and $\sqrt{\cdot}$ is taken elementwise. $\square$

### B.7 Proof of Theorem 3.8

*Proof.* Let a full SVD of $X^\top H^\star$ be given by $U\operatorname{Diag}(\sigma(X^\top H^\star))V^\top$ where $U \in \mathbb{R}^{d\times d}$, $V \in \mathbb{R}^{s\times s}$ are both orthogonal and $\sigma(X^\top H^\star) \in \mathbb{R}^{\min(d,s)} = \mathbb{R}^s$, since we assumed $s \leq \operatorname{rank}(X) \leq \min(N,d) \leq d$. Let $\tilde{U}$ be $U$ after dropping the last $d-s$ columns, we can then equivalently write

$$X^\top H^\star = \tilde{U}\overline{\operatorname{Diag}}(\sigma(X^\top H^\star))V^\top \tag{11}$$

where $\overline{\mathrm{Diag}}(\sigma(X^\top H^\star)) \in \mathbb{R}^{s \times s}$ is now a square diagonal matrix with $\sigma(X^\top H^\star)$ on the diagonal.

Since $H^\star$ is a solution of (D) by assumption, we know that there exists a $W^\star \in \partial G^*(X^\top H^\star)$ which is a solution of (P) by Proposition 2.3 and Assumption 2.4. By Proposition 3.10, this solution has the form $\tilde{U}\overline{\mathrm{Diag}}(\mu)V^\top$ where $\mu \in \partial g^*(\sigma(X^\top H^\star))$ where we again omitted the columns of $U$ that are multiplied with zero.

Therefore, it is required to calculate

$$W^{\star\top}\tilde{x} = V\overline{\mathrm{Diag}}(\mu)\tilde{U}^\top \tilde{x}. \tag{12}$$

This is however not yet kernelizable. To this end, we first note that (11) is equivalent to

$$\tilde{U} = X^\top H^\star V (\overline{\mathrm{Diag}}(\sigma(X^\top H^\star)))^{-1}$$

where we used the assumption that $H^{\star\top} K H^\star$ is nonsingular so that the singular values of $X^\top H^\star$ are nonzero. Secondly, we have an eigendecomposition

$$H^{\star\top} K H^\star = V\overline{\mathrm{Diag}}(\lambda)V^\top = V(\overline{\mathrm{Diag}}(\sigma(X^\top H^\star)))^2 V^\top$$

such that $\overline{\mathrm{Diag}}(\sigma(X^\top H^\star)) = \overline{\mathrm{Diag}}(\lambda)^{1/2}$ where $\lambda$ is a shorthand for $\lambda(H^{\star\top} K H^\star)$.

Using these two insights and (12), we find the desired result

$$W^{\star\top}\tilde{x} = V\overline{\mathrm{Diag}}(\mu \odot \lambda^{-1/2})V^\top H^{\star\top} X\tilde{x}$$

where $\mu \in \partial g^*(\sqrt{\lambda})$ and both $\sqrt{\lambda}$ and $\lambda^{-1/2}$ are elementwise. $\qquad\square$

### B.8 Proof of Theorem 3.11

*Proof.* Without loss of generality, we consider the statement for $X^\top X$. (DCA) for (d) and simultaneous iteration are shown in Algorithms 7 and 8 respectively. Here, $\overline{\mathbf{QR}}$ denotes a compact QR decomposition. The result then follows from noting that $\overline{\mathbf{QR}}$ and $\overline{\mathbf{SVD}}$ compute orthonormal bases for the column spaces of their arguments in $U$ and $Q$ respectively. Moreover, if $U^{(k)}$ and $Q^{(k)}$ span the same space and $V^{(k)} \in \mathbb{R}^{s \times r}$ has full rank, then we also have $\mathrm{col}(X^\top X U^{(k)} V^{(k)\top}) = \mathrm{col}(X^\top X Q^{(k)})$ where $\mathrm{col}(A)$ denotes the column space of $A$. Therefore, $U^{(k)}$ and $Q^{(k)}$ form orthonormal bases for the same subspaces at every iteration and are related through orthogonal transformation by [37, Theorem 2.1.18]. $\qquad\square$

---

| **Algorithm 7** (DCA) for (d) | **Algorithm 8** Simultaneous iteration for $X^\top X$ |
|---|---|
| **Require:** Matrix $X \in \mathbb{R}^{N \times d}$ | **Require:** Matrix $X \in \mathbb{R}^{N \times d}$ |
| 1: Initialize $W^{(0)} \in \mathbb{R}^{d \times s}$ | 1: Initialize $W^{(0)} \in \mathbb{R}^{d \times s}$ |
| 2: Initialize $k \leftarrow 0$ | 2: Initialize $k \leftarrow 0$ |
| 3: **repeat** | 3: **repeat** |
| 4: $\quad [U^{(k)}, \Sigma^{(k)}, V^{(k)}] \leftarrow \overline{\mathbf{SVD}}(X^\top X W^{(k)})$ | 4: $\quad [Q^{(k)}, R^{(k)}] \leftarrow \overline{\mathbf{QR}}(X^\top X W^{(k)})$ |
| 5: $\quad W^{(k+1)} \leftarrow U^{(k)} V^{(k)\top}$ | 5: $\quad W^{(k+1)} \leftarrow Q^{(k)}$ |
| 6: $\quad k \leftarrow k + 1$ | 6: $\quad k \leftarrow k + 1$ |
| 7: **until** convergence | 7: **until** convergence |

---

### B.9 Proof of Proposition 4.1

*Proof.* First, the objective from (c) without the square can be written as

$$\operatorname*{minimize}_{\substack{W \in \mathbb{R}^{d \times s} \\ W^\top W = I_s}} \sum_{i=1}^{N} \sqrt{(X_{i,:} - WW^\top X_{i,:})^\top (X_{i,:} - WW^\top X_{i,:})}$$

which is equivalent to

$$\operatorname*{minimize}_{\substack{W \in \mathbb{R}^{d \times s} \\ W^\top W = I_s}} \sum_{i=1}^{N} \sqrt{\|X_{i,:}\|^2 - \|W^\top X_{i,:}\|^2}$$

where we expanded and used the constraint. This minimization problem can be written as

$$\underset{\substack{W \in \mathbb{R}^{d \times s} \\ W^\top W = I_s}}{\text{minimize}} \; -\sum_{i=1}^{N} f_i((XW)_{i,:})$$

where

$$f_i : \mathbb{R}^s \to \overline{\mathbb{R}} : w \mapsto \begin{cases} -\sqrt{\|X_{i,:}\|^2 - \|w\|^2} & \text{if } \|w\| \le \|X_{i,:}\|, \\ +\infty & \text{otherwise,} \end{cases}$$

which is a convex function (see [61, p. 106]). The separable sum of convex functions is again convex and the objective is therefore to maximize a convex function. By the same reasoning as in Section 3.2, the constraint set can again be relaxed to its convex hull, being the spectral norm unit ball. The result now follows from taking $G = \delta_{B_{\|\cdot\|_{S_\infty}}}$ and $F = \sum_{i=1}^{N} f_i((\cdot)_{i,:})$, where the conjugates follow by (5), and combining (4) with Fact A.2. $\qquad\square$

## C  DC algorithms

This Appendix is devoted to deriving the many algorithms obtained from (DCA). To this end, we first recall some known subdifferential results.

**Proposition C.1.** *Let* $F = \|\cdot\|_{S_1}$*, then*

$$\tilde{U}\tilde{V}^\top \in \partial F(X) = \{U \operatorname{Diag}(\mu)V^\top \mid \mu \in \partial\|\sigma(X)\|_1, U \operatorname{Diag}(\sigma(X))V^\top = \mathbf{SVD}(X)\}$$

*where* $\|\cdot\|_1$ *denotes the* $l_1$*-norm and* $\tilde{U}\tilde{\Sigma}\tilde{V}^\top$ *is a compact SVD of* $X$*.*

*Proof.* The expression of the subdifferential follows immediately from Proposition 3.10 and the definition of the Schatten 1-norm. To show that $\tilde{U}\tilde{V}^\top$ is a subgradient at $X$, from [61, Theorem 23.8] we have that

$$\partial\|\sigma(X)\|_1 = \partial|\sigma(X)_1| \times \partial|\sigma(X)_2| \times \cdots \times \partial|\sigma(X)_{\min(N,d)}|$$

where we assume that $X \in \mathbb{R}^{N \times d}$. Moreover, recall that

$$\partial|x| = \begin{cases} \{1\} & \text{if } x > 0, \\ [-1, 1] & \text{if } x = 0, \\ \{-1\} & \text{if } x < 0. \end{cases}$$

Therefore $\mu = \operatorname{sign}(\sigma(X)) \in \partial\|\sigma(X)\|_1$ where $\operatorname{sign}(x)$ is the elementwise sign function with the convention $\operatorname{sign}(0) = 0$. For this choice of $\mu$, we obtain

$$\tilde{U}\tilde{V}^\top = U \operatorname{Diag}(\mu)V^\top \in \partial F(X)$$

since the singular values are always nonnegative and the columns/rows of $U, V$ corresponding to zero singular values are multiplied with zero in the sum. $\qquad\square$

**Remark C.2.** It should be noted that when one of the singular values is zero, it is possible to choose any value in the interval $[-1, 1]$ for the corresponding entry in $\mu$. While this is generally not encountered in practice, it could lead to some interesting algorithms to deal with degenerate cases.

**Proposition C.3.** *Let* $F = \frac{1}{2}\|\cdot\|_{S_2}^2$*, then* $\partial F(X) = \{X\}$*.*

*Proof.* The statement follows from the fact that $F$ is differentiable and [8, Theorem 3.33]. $\qquad\square$

**Proposition C.4.** *Let* $F = \|\cdot\|_{S_2}$*, then* $\frac{X}{\|X\|_{S_2}} \in \partial F(X)$ *for* $X \neq 0$*.*

*Proof.* The result follows immediately from Proposition 3.10 and the fact that $\partial\|x\|_2 = \{\frac{x}{\|x\|_2}\}$ for $x \neq 0$ since it is differentiable in that case. $\qquad\square$

**Remark C.5.** The case $X = 0$ in the preceding Proposition can be handled by using the fact that $\partial\|0\|_2 = B_{\|\cdot\|_2}$.

**Proposition C.6.** *Let $F = \frac{1}{2}\|\cdot\|_{S_1}^2$, then $\operatorname{tr}(\tilde{\Sigma})\tilde{U}\tilde{V}^\top = \|X\|_{S_1}\tilde{U}\tilde{V}^\top \in \partial F(X)$, where $\tilde{U}\tilde{\Sigma}\tilde{V}^\top$ is a compact SVD of $X$.*

*Proof.* Note that $F = g \circ H$ where $g(t) = \frac{1}{2}\max(0,t)^2$ is a nondecreasing, differentiable convex function and $H = \|\cdot\|_{S_1}$ is convex. By the chain rule of subdifferential calculus [8, Theorem 3.47], we have that $\partial F(X) = g'(H(X))\partial H(X)$. The desired subgradient then follows after using Proposition C.1. $\square$

**Proposition C.7.** *Let $F = \frac{3}{4}\|\cdot\|_{S_{4/3}}^{4/3}$, then $\partial F(X) = \{\tilde{U}\tilde{\Sigma}^{1/3}\tilde{V}^\top\}$, where $\tilde{U}\tilde{\Sigma}\tilde{V}^\top$ is a compact SVD of $X$.*

*Proof.* Note that $F = \frac{3}{4}\|\cdot\|_{4/3}^{4/3} \circ \sigma$ where $\|\cdot\|_{4/3}$ denotes the $l_{4/3}$-norm

$$\|\cdot\|_{4/3} : \mathbb{R}^q \to \mathbb{R} : x \mapsto (|x_1|^{4/3} + |x_2|^{4/3} + \cdots + |x_q|^{4/3})^{3/4}.$$

Moreover, we have that the gradient of $\frac{3}{4}\|\cdot\|_{l_{4/3}}^{4/3}$ is given by the elementwise cube root of its argument. The desired result then follows from Proposition 3.10 and the same arguments from the proof of Proposition C.1 to use a compact SVD instead of a full SVD. $\square$

## C.1 Derivation of Algorithms 1 and 2

*Proof.* The formulation (l) (which is equivalent to (d) after relaxing the constraint), can be written as (P) with $G = \delta_{B_{\|\cdot\|_{S_\infty}}}$ and $F = \frac{1}{2}\|\cdot\|_{S_2}^2$. Algorithm 1 then follows from (DCA) by noting that $\partial F(X) = \{X\}$ (Proposition C.3) and $\tilde{U}\tilde{V}^\top \in \partial G^*(Y)$ where $\tilde{U}\tilde{\Sigma}\tilde{V}^\top = \overline{\mathbf{SVD}}(Y)$ by (5) and Proposition C.1. The derivation of Algorithm 2 follows analogously. $\square$

## C.2 Derivation of Algorithm 3

*Proof.* The primal formulation of Proposition 3.3 can be written as (P) with $G = \frac{1}{4}\|\cdot\|_{S_4}^4$ and $F = \frac{1}{2}\|\cdot\|_{S_2}^2$. Algorithm 3 then follows from (DCA) by using $\partial F(X) = \{X\}$ (Proposition C.3) and $\partial G^*(Y) = \{\tilde{U}\tilde{\Sigma}^{1/3}\tilde{V}^\top\}$ where $\tilde{U}\tilde{\Sigma}\tilde{V}^\top = \overline{\mathbf{SVD}}(Y)$ by (8) and Proposition C.7. $\square$

## C.3 Derivation of Algorithm 4

*Proof.* The dual formulation of Proposition 3.3 can be written as (P) with $F = \frac{3}{4}\|\cdot\|_{S_{4/3}}^{4/3}$, $G = \frac{1}{2}\|\cdot\|_{S_2}^2$ where $X^\top$ takes the role of $X$. Using (DCA), $\partial G^*(X) = \{X\}$ (Proposition C.3) and Proposition C.7 yields the updates

$$[U^{(k)}, \Sigma^{(k)}, V^{(k)\top}] \leftarrow \overline{\mathbf{SVD}}(X^\top H^{(k)}), \qquad H^{(k+1)} \leftarrow X U^{(k)}(\Sigma^{(k)})^{1/3} V^{(k)\top}.$$

Now note that

$$U^{(k)} = X^\top H^{(k)} V^{(k)} \Sigma^{(k)-1}$$

where $\Sigma^{(k)}$ is invertible since a compact SVD is taken. Moreover, a compact eigendecomposition of $H^{(k)\top} K H^{(k)}$ yields $V^{(k)}\Lambda^{(k)}V^{(k)\top} = H^{(k)\top} K H^{(k)}$ where $\Lambda^{(k)} = \Sigma^{(k)2}$. The update is therefore equivalent to

$$[V^{(k)}, \Lambda^{(k)}] \leftarrow \overline{\mathbf{EIG}}(H^{(k)\top} K H^{(k)}), \qquad H^{(k+1)} \leftarrow K H^{(k)} V^{(k)} \Sigma^{(k)} \Lambda^{(k)-1/3} V^{(k)\top}$$

which is exactly Algorithm 4. $\square$

## C.4 Derivation of other DC algorithms

In this subsection, we consider kernelizable dual updates of (DCA) applied to the problems (l) through (q). First, we have the following general result when $G$ is unitarily invariant.

**Proposition C.8.** *Let $X \in \mathbb{R}^{N \times d}$ be the data matrix, $s \leq \operatorname{rank}(X)$ and define the kernel matrix $K = XX^\top$. Consider the primal problem (P) and suppose $G = g \circ \sigma$ is unitarily invariant. Assume $H^{(k)}{}^\top K H^{(k)}$ is invertible for each $k$, then the iterations of (DCA) applied to the dual are*

$$[V^{(k)}, \overline{\operatorname{Diag}}(\lambda^{(k)})] \leftarrow \overline{\mathbf{EIG}}(H^{(k)}{}^\top K H^{(k)})$$

$$H^{(k+1)} \in \partial F\left(K H^{(k)} V^{(k)} \overline{\operatorname{Diag}}(\mu^{(k)} \oslash \sqrt{\lambda^{(k)}}) V^{(k)}{}^\top\right)$$

*where $\mu^{(k)} \in \partial g^*(\sqrt{\lambda^{(k)}})$. We use the notation $\oslash$ and $\sqrt{\cdot}$ to denote elementwise division and square root respectively.*

*Proof.* The dual problem (D) is

$$\underset{H \in \mathbb{R}^{N \times s}}{\text{minimize}} \; F^*(H) - (g^* \circ \sigma)(X^\top H)$$

from Proposition 3.5. By using Proposition 3.10 (note that it contains a full SVD and not a compact SVD), the iterations of (DCA) applied to the dual problem are

$$[U^{(k)}, \operatorname{Diag}(\sigma^{(k)}), V^{(k)}{}^\top] \leftarrow \mathbf{SVD}(X^\top H^{(k)}), \quad H^{(k+1)} \leftarrow \partial F^{**}(X U^{(k)} \operatorname{Diag}(\mu^{(k)}) V^{(k)}{}^\top)$$

where $\mu^{(k)} \in \partial g^*(\sigma^{(k)})$. Since $H^{(k)}{}^\top K H^{(k)}$ is nonsingular, each entry of $\sigma^{(k)}$ is strictly greater than 0 and we may use a compact SVD and compact eigenvalue decomposition. The result then follows from the same arguments as in Subsection C.3 and the fact that $F^{**} = F^*$ [7, Cor. 13.38]. $\square$

**Remark C.9.** Instead of the assumption that $H^{(k)}{}^\top K H^{(k)}$ is invertible for each $k$, the result also holds whenever $y_i = 0$ implies that $0 \in \partial g^*(y)_i$ for each component and $0/0$ is taken to be 0. This is the case for all the functions encountered previously (Propositions C.1 through C.7).

Applying this result to the formulations (l) through (q) (or transposed variations) yields the updates in Table 4. It is clear that many of these algorithms are tightly related. For example, (14) is equivalent to Algorithm 2 as well as (16) since the scaling only affects $\Sigma^{(k)}$ which is discarded. Similarly, (18) is the same as (14) up to scaling. Moreover, (15) is (13) with additional normalization while (15) is also equivalent to (17) since the multiplication with the trace in (17) is void due to the normalization afterwards. In some sense, (DCA) is blind to simple problem transformations. Nevertheless, while they may be mathematically tightly related, the practical performance may differ significantly.

## C.5 Derivation of Algorithms 5 and 6

*Proof.* Recall from the proof of Proposition 4.1 that the primal problem can be written as (P) with $G = \delta_{B_{\|\cdot\|_{S_\infty}}}$ and $F = \sum_{i=1}^{N} f_i((\cdot)_{i,:})$. The expression for $\partial G^*$ follows immediately from (5) and Proposition C.1, while for $\partial F$ we use [61, Theorem 23.8] to find that the subdifferential of a separable sum is the Cartesian product of the subdifferentials. Moreover, we note that each $f_i$ can be written as $h_i \circ \|\cdot\|$ where

$$h_i : \mathbb{R} \to \overline{\mathbb{R}} : a \mapsto \begin{cases} -\sqrt{\|X_{i,:}\|^2 - a^2} & \text{if } |a| \leq \|X_{i,:}\|, \\ +\infty & \text{otherwise} \end{cases}$$

such that we can use Fact A.7. More concretely, we have that

$$\partial h_i(a) = \left\{ \frac{a}{\sqrt{\|X_{i,:}\|^2 - a^2}} \right\}$$

for $|a| < \|X_{i,:}\|$ and the empty set otherwise. Combining these subdifferentials with (DCA) yields Algorithm 5. Note that due to the initialization as well as the update rule for $W^{(k)}$, the spectral norm of $W^{(k)}$ is always less than 1 such that the argument of $\partial h_i$ has absolute value $|a| \leq \|X_{i,:}\|$ and the square root is always well-defined. To derive Algorithm 6, the same subdifferentials can be used in combination with the same arguments as in the derivation of Algorithm 4 to make it kernelizable. $\square$

**Remark C.10.** Whenever $(XW^{(k)})_{i,:} = 0$ in Algorithm 5, $Y_{i,:}^{(k)}$ can be set to an arbitrary vector in some ball of some radius, in accordance with the second case in Fact A.7. Another edge case occurs whenever $\|X_{i,:}\|^2 = \|(XW^{(k)})_{i,:}\|^2$. In practice, this is alleviated by adding some regularization constant $\varepsilon$ in the denominator. Formally, this is accomplished by modifying the first case in the definition of $h_i$ to be $-\sqrt{\|X_{i,:}\|^2 - a^2 + \varepsilon^2}$ for $|a| \leq \sqrt{\|X_{i,:}\|^2 + \varepsilon^2}$ as in [9].

Table 4: DC algorithms for formulations (l) through (q) where the data matrix is $X \in \mathbb{R}^{N \times d}$. Only the kernelizable versions involving $K := XX^\top$ are provided.

| Problem | (DCA) dual updates (kernelizable versions) | Derived through | |
|---|---|---|---|
| (l) | $[V^{(k)}, \Lambda^{(k)}] \leftarrow \overline{\mathbf{EIG}}(H^{(k)\top} K H^{(k)})$ 
 $H^{(k+1)} \leftarrow K H^{(k)} V^{(k)} \Lambda^{(k)-1/2} V^{(k)\top}$ | Proposition C.1 
 Proposition C.3 | (13) |
| (m) | $[U^{(k)}, \Sigma^{(k)}, V^{(k)\top}] \leftarrow \overline{\mathbf{SVD}}(K H^{(k)})$ 
 $H^{(k+1)} \leftarrow U^{(k)} V^{(k)\top}$ | Proposition C.3 
 Proposition C.1 | (14) |
| (n) | $[V^{(k)}, \Lambda^{(k)}] \leftarrow \overline{\mathbf{EIG}}(H^{(k)\top} K H^{(k)})$ 
 $\tilde{H}^{(k+1)} \leftarrow K H^{(k)} V^{(k)} \Lambda^{(k)-1/2} V^{(k)\top}$ 
 $H^{(k+1)} \leftarrow \dfrac{\tilde{H}^{(k+1)}}{\|\tilde{H}^{(k+1)}\|_{S_2}}$ | Proposition C.1 
 Proposition C.4 | (15) |
| (o) | $[U^{(k)}, \Sigma^{(k)}, V^{(k)\top}] \leftarrow \overline{\mathbf{SVD}}(\mathrm{tr}(H^{(k)\top} K H^{(k)})^{-1/2} K H^{(k)})$ 
 $H^{(k+1)} \leftarrow U^{(k)} V^{(k)\top}$ | Proposition C.4 
 Proposition C.1 | (16) |
| (p) | $[V^{(k)}, \Lambda^{(k)}] \leftarrow \overline{\mathbf{EIG}}(H^{(k)\top} K H^{(k)})$ 
 $\tilde{H}^{(k+1)} \leftarrow \mathrm{tr}(\Lambda^{(k)1/2}) K H^{(k)} V^{(k)} \Lambda^{(k)-1/2} V^{(k)\top}$ 
 $H^{(k+1)} \leftarrow \dfrac{\tilde{H}^{(k+1)}}{\|\tilde{H}^{(k+1)}\|_{S_2}}$ | Proposition C.6 
 Proposition C.4 | (17) |
| (q) | $[U^{(k)}, \Sigma^{(k)}, V^{(k)\top}] \leftarrow \overline{\mathbf{SVD}}(K H^{(k)})$ 
 $H^{(k+1)} \leftarrow \mathrm{tr}(H^{(k)\top} K H^{(k)})^{-1/2} \mathrm{tr}(\Sigma^{(k)}) U^{(k)} V^{(k)\top}$ | Proposition C.4 
 Proposition C.6 | (18) |

# D  IRLS and DCA

In this Appendix, we derive an iteratively reweighted least squares (IRLS) algorithm for the robust PCA formulation from Section 4 and compare with Algorithm 5. As is common in the literature (see [45] and references therein), the IRLS update for the robust subspace recovery problem is of the form

$$W^{(k+1)} \in \underset{\substack{W \in \mathbb{R}^{d \times s} \\ \|W\|_{S_\infty} \leq 1}}{\arg\min} \sum_{i=1}^{N} \frac{1}{\beta_i(W^{(k)})} \|X_{i,:} - WW^\top X_{i,:}\|^2$$

where

$$\beta_i(W^{(k)}) = \|X_{i,:} - W^{(k)} W^{(k)\top} X_{i,:}\|.$$

This minimization problem is equivalent to

$$W^{(k+1)} \in \underset{\substack{W \in \mathbb{R}^{d \times s} \\ \|W\|_{S_\infty} \leq 1}}{\arg\min} \|BX - WW^\top BX\|_{S_2}^2 = \underset{\substack{W \in \mathbb{R}^{d \times s} \\ \|W\|_{S_\infty} \leq 1}}{\arg\min} -\frac{1}{2} \|BXW\|_{S_2}^2$$

where $B$ is a square diagonal matrix with $(\beta_i(W^{(k)}))^{-1/2}$ on the diagonal. The latter is recognized to be (l) such that we may use Proposition 3.2 to obtain a solution of the inner problem. This leads to Algorithm 10, where we also used the fact that if $A = U\Sigma V^\top$ is a SVD of $A$, then $A^\top = V\Sigma^\top U^\top$ is a SVD of $A^\top$. Note that for each $k \geq 1$, $W^{(k)\top} W^{(k)} = I_s$ such that $\beta_i^{(k)}$ could equivalently be written as $\sqrt{\|X_{i,:}\|^2 - \|(XWp(k))\|_{i,:}^2}$, which is exactly the denominator on line 5 of Algorithm 9.

The main difference between the two algorithms is that an additional square root is taken of $\beta_i^{(k)}$ in Algorithm 10, while in Algorithm 9 a multiplication with $(XW^{(k)})_{i,:}$ is performed instead. Another difference is that an orthogonal factor remains in Algorithm 9, which is reminiscent of the connection between Algorithm 7 and simultaneous iteration, see Theorem 3.11.

**Algorithm 9** (DCA) for Proposition 4.1 (primal)

**Require:** Matrix $X \in \mathbb{R}^{N \times d}$
1: Initialize $W^{(0)} \in \mathbb{R}^{d \times s}$ s.t. $\|W^{(0)}\|_{S_\infty} = 1$
2: Initialize $k \leftarrow 0$
3: **repeat**
4:   **for** $i = 1$ **to** $N$ **do**
5:     $Y_{i,:}^{(k)} \leftarrow \dfrac{(XW^{(k)})_{i,:}}{\sqrt{\|X_{i,:}\|^2 - \|(XW^{(k)})_{i,:}\|^2}}$
6:   **end for**
7:   $[U^{(k)}, \Sigma^{(k)}, V^{(k)\top}] \leftarrow \overline{\mathbf{SVD}}(X^\top Y^{(k)})$
8:   $W^{(k+1)} \leftarrow U^{(k)} V^{(k)\top}$
9:   $k \leftarrow k + 1$
10: **until** convergence

**Algorithm 10** IRLS for robust subspace recovery

**Require:** Matrix $X \in \mathbb{R}^{N \times d}$
1: Initialize $W^{(0)} \in \mathbb{R}^{d \times s}$ s.t. $\|W^{(0)}\|_{S_\infty} = 1$
2: Initialize $k \leftarrow 0$
3: **repeat**
4:   **for** $i = 1$ **to** $N$ **do**
5:     $\beta_i^{(k)} \leftarrow \|X_{i,:} - W^{(k)}W^{(k)\top}X_{i,:}\|$
6:   **end for**
7:   $B^{(k)} \leftarrow \overline{\mathrm{Diag}}(\beta^{-1/2})$
8:   $[U^{(k)}, \Sigma^{(k)}, V^{(k)\top}] \leftarrow \overline{\mathbf{SVD}}(X^\top B^{(k)})$
9:   $W^{(k+1)} \leftarrow U_{:,1:s}^{(k)}$
10:   $k \leftarrow k + 1$
11: **until** convergence

## E    DC formulations for sparse PCA

The DC duality framework is very broad. We now discuss two promising avenues for sparse PCA.

First, similar to formulation (q), we can consider the DC formulation

$$\operatorname*{minimize}_{W \in \mathbb{R}^{d \times s}} \frac{1}{2}\|W\|_S^2 - \|XW\|_{S_2},$$

where $\|\cdot\|_S$ is a sparsity inducing norm such as taking the sum of the absolute value of its elements ($l_1$-norm on its vectorization). Note that this is not a classical sparse PCA formulation but achieves a similar purpose. The convex conjugate of this function follows from (1).

Second, if we consider only one principal component, then a DC formulation for sparse PCA is described in [9]. This formulation takes the form

$$\operatorname*{minimize}_{w \in \mathbb{R}^d} \delta_C(w) - \frac{1}{2}\|Xw\|_{S_2}^2,$$

where $C$ is the convex hull of the intersection of the unit norm ball and $\{w \in \mathbb{R}^d \mid \|w\|_0 \le k\}$. The conjugate of this indicator function can be computed in closed form. Indeed, $\delta_C^*(z) = \max_{w \in C} w^\top z$ by definition. Then, since $w$ has at most $k$ nonzeros, we see that this maximization problem is solved when the support of $w$ corresponds to the $k$ largest entries of $z$ in absolute value. Taking into account that $w$ is also a unit vector, it follows that $\delta_C^*(z)$ is exactly the Euclidean norm of the vector containing the $k$ largest entries of $z$ in absolute value.

Both of these formulations are not unitarily invariant and therefore not easily kernelizable through Proposition 3.5 (a sufficient condition for kernelizability). Though, to the best of our knowledge, sparse PCA is generally not used in the kernel setting since the main motivation for sparse PCA is interpretability of the data and principal components. More concretely, by restricting the cardinality of the principal components, this implicitly assumes that they are linear combinations of just a few innput variables, which does not mesh well with kernel methods that are inherently nonlinear.

## F    Additional experiments

In this Appendix, we provide further experimental results.

**Higher accuracy**    Timing results for higher target accuracies than those in Section 5 are presented in Table 5. The problem instances are generated in a similar manner. We observe that the Krylov-type methods are more suited than the first-order methods in this setting. Nevertheless, ZeroFPR applied to (n) still outperforms the Krylov methods for larger matrices. We observe that the (DCA) algorithms are quite slow, which is mostly due to the relatively expensive check of the stopping criterion at each iteration.

Table 5: Timing results for various methods applied to PCA formulations with higher accuracies. The problem setting $(N, d, s, \varepsilon)$ is the same as in Table 1. All timings are in milliseconds, and timings longer than 5 seconds are not displayed.

| Method | $(3000, 2000, 20, 10^{-5})$ | $(2000, 3000, 20, 10^{-5})$ | $(3500, 3500, 20, 10^{-5})$ |
|---|---|---|---|
| ZeroFPR (l) | $2009.0 \pm 122.5$ | $2202.6 \pm 296.0$ | $4269.3 \pm 57.6$ |
| ZeroFPR (n) | $735.0 \pm 295.7$ | $876.1 \pm 194.8$ | $1269.1 \pm 357.0$ |
| ZeroFPR (o) | $1092.5 \pm 285.1$ | $1532.8 \pm 213.1$ | $2577.7 \pm 274.3$ |
| ZeroFPR (q) | $2018.4 \pm 141.1$ | $2148.0 \pm 319.6$ | $4245.3 \pm 715.5$ |
| PG (n) | / | $3569.9 \pm 410.1$ | / |
| PG (o) | $4011.7 \pm 969.8$ | / | / |
| Algorithm 1 | $4709.5 \pm 3963.0$ | / | / |
| Algorithm 2 | / | $2559.5 \pm 832.2$ | / |
| Algorithm 3 | / | / | / |
| Algorithm 4 | / | / | / |
| SVDS | $625.3 \pm 10.6$ | $556.9 \pm 18.7$ | $1564.8 \pm 31.1$ |
| KrylovKit | $773.8 \pm 29.7$ | $696.0 \pm 17.5$ | $1858.7 \pm 24.3$ |

**Different spectra** The performance of Krylov methods for computing the top $s$ singular values is known to be highly sensitive to the spectrum, and in particular the spectral gaps $\sigma_i / \sigma_{i+1}$. Table 6 reports timing results for various methods applied to matrices with different fixed spectra. In the first column, where the singular values decay exponentially, all methods perform equally well. In the second column, the singular values decrease linearly from 500 to 100, resulting in small spectral gaps. Here, the Krylov methods are roughly 10 times slower compared to the exponential decay case. Notably, ZeroFPR and PG on the formulations (n) and (o) significantly outperform the Krylov solvers. This trend continues in the third column, where the singular values decrease linearly from 500 to $10^{-8}$.

Table 6: Timing results for various methods applied to PCA formulations for matrices with specified spectrum. All matrices are square with $N = 3500$. A stopping criterion tolerance of $10^{-3}$ was used to find $s = 20$ principal components. All timings are in milliseconds, and timings longer than 5 seconds are not displayed.

| Method | $\sigma_i = 100 \cdot 0.9^i$ | $\sigma_i = 500 - (i-1)\frac{400}{N-1}$ | $\sigma_i = 500 - (i-1)\frac{500-10^{-8}}{N-1}$ |
|---|---|---|---|
| ZeroFPR (l) | $490.7 \pm 123.5$ | / | / |
| ZeroFPR (n) | $171.9 \pm 64.8$ | $456.2 \pm 90.1$ | $455.7 \pm 80.0$ |
| ZeroFPR (o) | $260.1 \pm 68.1$ | $1043.3 \pm 164.0$ | $1137.7 \pm 165.6$ |
| ZeroFPR (q) | $298.5 \pm 97.2$ | / | / |
| PG (n) | $115.8 \pm 22.1$ | $730.2 \pm 55.0$ | $729.5 \pm 71.1$ |
| PG (o) | $262.1 \pm 40.3$ | $1068.8 \pm 34.3$ | $1173.2 \pm 88.5$ |
| Algorithm 1 | $300.2 \pm 13.9$ | $3633.2 \pm 184.8$ | $3581.2 \pm 554.2$ |
| Algorithm 2 | $295.7 \pm 14.5$ | $3594.1 \pm 36.0$ | $3866.0 \pm 59.4$ |
| Algorithm 3 | $429.7 \pm 19.2$ | / | / |
| Algorithm 4 | $387.1 \pm 16.2$ | / | / |
| SVDS | $288.9 \pm 2.1$ | $3113.9 \pm 133.4$ | $2588.5 \pm 178.1$ |
| KrylovKit | $242.8 \pm 3.1$ | $4250.3 \pm 392.9$ | $3653.5 \pm 309.8$ |

**Real-world datasets** Table 7 shows the timing results of the best performing methods on the MNIST dataset [22] ($N = 60000, d = 784$) with a tolerance of $\varepsilon = 10^{-3}$. We observe the same behavior as in the synthetic experiments: the generic first-order methods are faster than the classical eigensolvers in this setting.

Table 8 shows the timing results of the best performing methods on the 100k top words from the 2024 Wikipedia + Gigaword 5, 50d GloVe word embedding dataset [57, 16] ($N = 100000, d = 50$) with a tolerance of $\varepsilon = 10^{-3}$. While the first-order methods are still faster than the classical eigensolvers,

this difference is less pronounced. We attribute this result to the fact that these word embeddings already form a compressed version of the word vector space, and therefore have favorable spectral characteristics, similar to the results encountered in the first column of Table 6.

Table 7: Timing results for different methods of applying PCA to the MNIST dataset with a tolerance of $\varepsilon = 10^{-3}$. The column headers denote the number of principal components. All timings are in milliseconds, and timings longer than 5 seconds are not displayed.

| Method | 30 | 50 | 100 | 150 |
|---|---|---|---|---|
| ZeroFPR (n) | $286.7 \pm 92.2$ | $444.7 \pm 86.1$ | $1734.4 \pm 247.0$ | $2363.3 \pm 135.2$ |
| PG (n) | $215.5 \pm 35.7$ | $351.9 \pm 62.4$ | $1254.8 \pm 103.1$ | $1772.2 \pm 150.5$ |
| SVDS | $1685.1 \pm 226.2$ | $2576.5 \pm 4.5$ | / | / |

Table 8: Timing results for different methods of applying PCA to the GloVe word embedding dataset with a tolerance of $\varepsilon = 10^{-3}$. The column headers denote the number of principal components. All timings are in milliseconds.

| Method | 10 | 15 | 20 | 30 |
|---|---|---|---|---|
| ZeroFPR (n) | $131.1 \pm 114.5$ | $208.2 \pm 150.4$ | $271.8 \pm 147.2$ | $367.1 \pm 198.2$ |
| PG (n) | $76.0 \pm 71.4$ | $120.0 \pm 100.4$ | $165.3 \pm 111.9$ | $250.4 \pm 113.2$ |
| SVDS | $200.6 \pm 14.6$ | $281.0 \pm 31.4$ | $319.7 \pm 11.5$ | $329.0 \pm 34.5$ |

**Additional formulations**    In Table 9, we present some timings for the algorithms described in Table 4, where we used the relative error with respect to the optimal value for the corresponding formulation as a stopping criterion. Wherever possible, we reuse information from the SVDs/EIGs to cheaply evaluate the objective value. We observe that in general, the algorithms using eigenvalue decompositions are faster than those involving singular value decompositions. Moreover, the algorithms are on par with the Krylov solvers whenever the number of features is greater than or equal to the number of datapoints. This is to be expected since the algorithms use the kernel matrix. In the opposite setting, one should use algorithms using the (scaled) covariance matrix instead.

Table 9: Timing results for methods from Table 4 applied to PCA formulations. The problem setting $(N, d, s, \varepsilon)$ is the same as in Table 1. All timings are in milliseconds.

| Method | $(3000, 2000, 20, 10^{-3})$ | $(2000, 3000, 20, 10^{-3})$ | $(3000, 3000, 20, 10^{-3})$ |
|---|---|---|---|
| Iteration (13) | $1135.6 \pm 72.7$ | $520.4 \pm 58.3$ | $1198.9 \pm 153.3$ |
| Iteration (14) | $1327.5 \pm 331.8$ | $567.6 \pm 78.6$ | $1179.1 \pm 272.4$ |
| Iteration (15) | $792.4 \pm 88.1$ | $338.1 \pm 42.3$ | $800.7 \pm 79.9$ |
| Iteration (16) | $1059.4 \pm 65.5$ | $556.6 \pm 58.7$ | $1182.2 \pm 94.0$ |
| Iteration (17) | $1742.6 \pm 206.4$ | $774.6 \pm 84.4$ | $1815.1 \pm 71.7$ |
| Iteration (18) | $5926.6 \pm 149.6$ | $1085.7 \pm 76.0$ | $4895.1 \pm 94.9$ |
| SVDS | $638.1 \pm 41.7$ | $522.3 \pm 44.4$ | $1164.0 \pm 57.5$ |
| KrylovKit | $755.5 \pm 26.3$ | $685.4 \pm 44.3$ | $1215.7 \pm 43.5$ |

**A note on tolerance**    Previous comparisons used a fixed tolerance $\varepsilon$ across methods, but this is not ideal since (1) the problem formulations differ, and (2) algorithms interpret $\varepsilon$ differently in their stopping criteria. To see this impact, we constructed 10 different $50 \times 50$ matrices with entries sampled from the standard normal distribution. Then, for each tolerance from $10^{-1}$ down to $10^{-9}$ on a log-spaced grid, we computed $s = 10$ components, and evaluated the relative error with respect to the optimal solution on the formulations (n) or (o). We averaged these errors over the 10 different matrices and present the results in Figure 4. We observe that KrylovKit.jl reaches machine precision for the function value with $\varepsilon = 10^{-3}$ while ZeroFPR and PG perform the worst on (n) and (o).

**Robust (kernel) PCA**    Consider the MNIST dataset [22] with a train-test split of 80-20. To verify the robustness properties of the robust PCA formulation in Section 4, we contaminate 15% of the

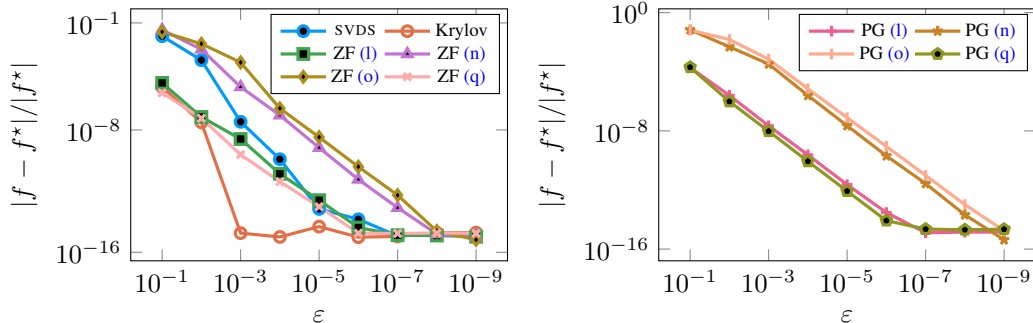

Figure 4: Relative error of the objective value (n) or (o) for multiple methods with varying tolerance for the stopping error criterion. $f^\star$ denotes the optimal value while ZF is shorthand for ZeroFPR.

training data with heavy Gaussian noise ($\sigma = 15$) and leave the test set untouched. We consider four different settings:

1. Linear PCA on the noncontaminated data (baseline)
2. Linear PCA on the contaminated data
3. Robust PCA on the contaminated data using Algorithm 5
4. Robust PCA on the contaminated data using Algorithm 6 with a linear kernel (note that the kernel matrix does not fit in memory but we can use its factored representation since the kernel is linear)

For each of these settings, we evaluate the reconstruction error on the noncontaminated test set. These errors are summarized in Table 10. The robust methods outperform standard linear PCA in the contaminated setting. Further, the similar performance of the two robust methods is to be expected, due to strong duality. Lastly, the top components are less affected by the outliers than the remaining components. This is logical since these latter components explain less 'variance' and it becomes more difficult to distinguish noise/outliers from data.

Table 10: Reconstruction errors on the test set of various PCA schemes. The column headers denote the number of principal components. The robust PCA formulations outperform standard linear PCA on the contaminated dataset.

| Method | 50 | 100 | 150 |
|---|---|---|---|
| 1. Linear PCA (noncontaminated) | 0.011 | 0.0057 | 0.0034 |
| 2. Linear PCA (contaminated) | 0.062 | 0.0569 | 0.0520 |
| 3. Robust PCA (contaminated) with Algorithm 5 | 0.014 | 0.0107 | 0.0090 |
| 4. Robust PCA (contaminated) linear kernel with Algorithm 6 | 0.014 | 0.0107 | 0.0090 |

To further illustrate the strength and usecase of this formulation, we can now extract robust features in light of Theorem 3.8. To this end, for each of the settings, we train a small multilayer perceptron classifier (1 hidden layer with 20 neurons) on these extracted features. Note that we choose a very simple classifier so the quality of the features becomes more apparent. Additionally, we also consider the following two settings:

5. Robust PCA on the noncontaminated data using Algorithm 5
6. Robust kernel PCA on the contaminated data using Algorithm 6 with a RBF kernel (length scale paramter $\gamma = 0.01$) approximated using a Nyström approximation with 500 pivots

The test accuracies of each classifier are displayed in Table 11. We observe that the robust formulations always perform better than the non-robust formulations for the contaminated data. The robust formulation on the noncontaminated data (setting 5) also does not degrade in performance with respect to the baseline (setting 1). Lastly, the RBF kernel trained on the contaminated data performs on par with models trained on uncontaminated data.

Table 11: Test accuracies of a small multilayer perceptron after feature extraction.

| Method | 50 | 100 | 150 |
|---|---|---|---|
| 1. Linear PCA (noncontaminated) | 95.87 | 95.73 | 94.84 |
| 2. Linear PCA (contaminated) | 76.57 | 79.93 | 81.09 |
| 3. Robust PCA (contaminated) with Algorithm 5 | 86.89 | 87.32 | 85.53 |
| 4. Robust PCA (contaminated) linear kernel with Algorithm 6 | 87.63 | 84.44 | 86.92 |
| 5. Robust PCA (noncontaminated) with Algorithm 5 | 95.73 | 96.06 | 95.43 |
| 6. Robust PCA (contaminated) RBF kernel with Algorithm 6 | 95.21 | 95.43 | 94.34 |

As a last experiment, we compare the robust PCA formulation with the classical robust PCA [15]. Note that the two approaches are of a different nature. The classical method decomposes a matrix into a low-rank component and a sparse component, which is well-suited for motion segmentation. In contrast, the formulation in Section 4 is based on the autoencoder perspective and aims to enforce sparsity on the reconstruction errors, which is better aligned with outlier detection. Another major difference is that the classical method is not out-of-sample applicable while ours is through Theorem 3.8. These differences make fair comparisons difficult.

To further illustrate this fact, consider the synthetic experiment from [15, Section 4.1]. We first generate a low-rank component $L^\star$ according to the same settings and add a sparse matrix $E$ where $E$ has limited support, with the nonzero entries of $E$ being independent Bernoulli $\pm 1$ entries. We consider two settings:

1. The support of $E$ is chosen uniformly distributed over all its entries
2. The support of $E$ consists exclusively of full rows (i.e., an image where complete rows are perturbed)

We then compare the classical principal component pursuit (PCP) with alternating directions [15, Algorithm 1] with Algorithm 5. For our experiment, we choose $\text{rank}(L^\star) = 25$ and $L^\star \in \mathbb{R}^{500 \times 500}$. We assume the size of the support of $E$ is $|E| = 12500$ and compute both the reconstruction error of the low-rank component and the primal cost of Proposition 4.1 (i.e., the $l_1$-norm of the row-wise reconstruction errors) and summarize our results in Table 12 (all metrics were averaged over 20 different random problems). We observe that the classical PCP performs better in the setting where it was proposed while our algorithm is better in the outlier setting. Moreover, we see that the primal cost is lower for our algorithm in both cases, which is to be expected since the PCP algorithm is not designed to minimize this cost.

Table 12: Comparison principal component pursuit (PCP) and Algorithm 5 for decomposing a matrix into a low-rank component $L$ and a sparse component $E$.

| Setting | Method | $\frac{\|L - L^\star\|_{S_2}}{\|L^\star\|_{S_2}}$ | Primal cost |
|---|---|---|---|
| Uniform perturbation | PCP | $3 \times 10^{-8}$ | 2488 |
| | Algorithm 5 | 0.013 | 2364 |
| Full row perturbations | PCP | 0.045 | 559 |
| | Algorithm 5 | 0.009 | 545 |

