# OpenReview forum: "Rethinking PCA Through Duality"
_NeurIPS.cc/2025/Conference — NeurIPS 2025 poster_

### Official Review · Reviewer_A5aE · 2025-06-27

**Clarity:** 2
**Significance:** 3
**Originality:** 3
**Rating:** 4
**Confidence:** 2

**Summary:**

The paper aims at proposing a faster algorithm in solving PCA.

**Questions:**

could the authors comment on practical use  for the proposed algorithms?

**Ethical Concerns:**

["NO or VERY MINOR ethics concerns only"]

**Final Justification:**

Thanks for authors for their rebuttal.
I have updated my evaluation.

**Limitations:**

yes

**Quality:**

3

**Strengths And Weaknesses:**

**strengths:**
- thorough mathematical presentation of the proposed algorithms

**weaknesses:**
- paper difficult to follow in particular its purpose, goal and its  contributions e.g.  what are practical uses and potential impact of the proposed algorithm(s) for cited applications (e.g. transformer [64])?
- lines 108/176/311 missing Fig 1 and/or Fig 2. reference in the sentences ?
- Appendices are missing (?)
- Table 1 shows experimental results for computational times but it is difficult to interpret as different units are used (e.g. second, milli seconds), and it is unclear if they are statistically different.
- Tab1 considers three problem settings with variations only on variable $N$ (number of samples) and $d$ (feature dimension) while $s$ and $\epsilon$ remains the same. Would plots be  more suited to highlight impacts of $N$ and $d$ on compute  time?

---

> ### Author Rebuttal · Authors · 2025-07-31
>
> We thank the reviewer for their time and comments.
>
> ---
>
> In the following we address the stated **Weaknesses**.
>
> > paper difficult to follow in particular its purpose, goal and its contributions e.g. what are practical uses and potential impact of the proposed algorithm(s) for cited applications (e.g. transformer [64])?
>
> We recognize that the paper may not be the easiest to follow since the main contributions of the paper are of theoretical nature. However, we have motivated all our results clearly and made sure to state all the assumptions explicitly, while most mathematical details are deferred to the appendix. The main contributions are also stated in a separate section.
>
> We reiterate the main goals/contributions of this paper:
> - Developing novel (and useful) formulations for PCA. Note that it is quite remarkable that for such a well-studied and fundamental problem as PCA, there still exist problem formulations that have not been described before.
> - Discovering new connections between PCA, optimization, and numerical linear algebra.
> - Deriving results for a family of PCA like formulations that are kernelizable and out-of-sample applicable.
> - As an example of this last result, we consider a robust PCA formulation that minimizes the $l_1$-deviation of the reconstruction errors, thereby making the principal components less sensitive to outliers.
>
> Some examples of immediate practical applications of this paper are:
> - Faster PCA solvers for the accuracy required in machine learning applications. Note that PCA is still prevalent in many domains, e.g., in bio-informatics.
> - A way to robustly extract features in a kernel setting, with a strong theoretical foundation.
>
> The most promising research avenues that we have in mind for the future are:
> - Tailored algorithms for solving the novel problem formulations, thereby obtaining even faster solvers for PCA. Moreover, from the connection between kernel PCA and self-attention [1], these algorithms may inspire better algorithms for training LLMs.
> - Novel (deep) architectures inspired by the new formulations (see also our response to **Reviewer BYYV**).
> - Deriving novel theoretical results by utilizing the connection between the various scientific fields. In particular, since simultaneous iteration is known to converge linearly from numerical linear algebra [2, Theorem 28.4], this means that the corresponding difference-of-convex formulation must satisfy some growth condition. By utilizing this connection, we believe it is possible to reverse-engineer this condition, which may subsequently also hold for many other formulations, and so on.
>
> > lines 108/176/311 missing Fig 1 and/or Fig 2. reference in the sentences ?
>
> While the tags are properly hyperlinked to the corresponding location, we will make this more clear by giving hyperlinks a distinct color as well as mentioning the explicit Figure numbers.
>
> > Appendices are missing (?)
>
> The appendices are present in the supplementary material, as is allowed by the NeurIPS formatting guidelines: "Technical appendices with additional results, figures, graphs and proofs may be submitted with the paper submission before the full submission deadline (see above), or as a separate PDF in the ZIP file below before the supplementary material deadline."
>
> > Table 1 shows experimental results for computational times but it is difficult to interpret as different units are used (e.g. second, milli seconds), and it is unclear if they are statistically different.
>
> We will report all computation times in milliseconds to avoid ambiguity, as has already been done in the supplied Appendix. As for the statistical differences of the timings, we emphasize that we use BenchmarkTools.jl [3], which, to quote: "is robust in the presence of timer errors, OS jitter and other environmental fluctuations, and is insensitive to the highly nonideal statistics produced by timing measurements", thereby providing timing data that is more statistically reliable than what is typically encountered in many machine learning papers. In particular, we draw attention to Tables 6 and 7, for which it should be clear that the timings between ZeroFPR/PG and SVDS/KrylovKit are statistically different.
>
> > Tab1 considers three problem settings with variations only on variable $N$ (number of samples) and $d$ (feature dimension) while $s$ and $\varepsilon$ remains the same. Would plots be more suited to highlight impacts of $N$ and $d$ on compute time?
>
> We thank the reviewer for this recommendation. We have avoided plots for these results since they are quite cluttered. However, if space permits, we will include visualizations of only the best performing algorithms to better highlight the impacts of $N$ and $d$. See our response to **Reviewer BYYV** for some additional experiments with varying $s$ and see Table 5 in the appendix for an additional experiment with different $\varepsilon$.
>
> ---
>
> Regarding the posed **Question** concerning the practical use, see our response to the first **Weakness**.
>
> ---
>
> We would like to kindly request the reviewer to reconsider their score based on the other reviews and our rebuttal, since it is not clear how the stated weaknesses merit the current score.
>
> ---
>
> References
>
> [1] Teo, Rachel SY, and Tan Nguyen. Unveiling the hidden structure of self-attention via kernel principal component analysis. Advances in Neural Information Processing Systems 37 (2024): 101393-101427.
>
> [2] Trefethen, Lloyd N., and David Bau. Numerical linear algebra. Society for Industrial and Applied Mathematics, (2022).
>
> [3] Chen, Jiahao, and Jarrett Revels. Robust benchmarking in noisy environments. arXiv preprint arXiv:1608.04295 (2016).

---

### Official Review · Reviewer_jssA · 2025-07-01

**Clarity:** 3
**Significance:** 4
**Originality:** 4
**Rating:** 5
**Confidence:** 2

**Summary:**

This paper revisits Principal Component Analysis (PCA) through the framework of difference-of-convex (DC) duality, offering new theoretical insights and algorithmic developments. The authors establish a novel connection between PCA and DC optimization, showing that applying the Difference-of-Convex Algorithm (DCA) to variance maximization is closely related to simultaneous iteration, a classical numerical linear algebra technique akin to the QR algorithm. This reveals an unexpected link between optimization and traditional matrix computations. Additionally, the paper introduces a kernelizable dual formulation for PCA that supports out-of-sample extensions, making it suitable for modern machine learning applications. A key contribution is a new robust variant of kernel PCA that minimizes the $l_1$-norm deviation of reconstruction errors, along with an associated algorithm connected to iteratively reweighted least squares. Empirical results demonstrate that simple first-order optimization methods can outperform state-of-the-art solvers for certain formulations, depending on the required accuracy. The work opens several promising research directions, including exploring other unitarily invariant objectives, developing deep stacked (kernel) PCA architectures, and designing specialized algorithms for the proposed DC dual pairs. Overall, this study provides a fresh optimization-based perspective on PCA, bridging classical techniques with modern machine learning applications.

**Questions:**

1. Proposition 3.3 appears to rely on the distinctness of singular values. Could this assumption be relaxed while still preserving the validity of the proposition? Alternatively, does the non-distinct case lead to fundamentally different results?
2. The paper introduces several novel optimization-based algorithms for PCA. Could you provide a more rigorous discussion of their convergence properties? Specifically: Under what conditions (e.g., step sizes, initialization) do they converge? Are there local vs. global convergence guarantees? How does their convergence rate compare to classical methods? For the robust ℓ₁ variant, does the connection to IRLS imply any inherited convergence guarantees?
3. While Assumption 3.1 appears natural from a data processing standpoint, it represents a significant restriction from the perspective of numerical linear algebra where such constraints aren't typically required. If we relax this assumption, which of the proposed PCA reformulations remain valid?

**Ethical Concerns:**

["NO or VERY MINOR ethics concerns only"]

**Final Justification:**

I appreciate the theoretical contributions of the paper and continue to support it. My rating remains unchanged.

**Limitations:**

yes

**Paper Formatting Concerns:**

No concern.

**Quality:**

4

**Strengths And Weaknesses:**

Strengths:
The paper provides a fresh perspective on PCA by framing it within difference-of-convex (DC) duality, uncovering new connections between optimization and classical linear algebra. The link between the DC algorithm (DCA) and simultaneous iteration (related to the QR algorithm) is particularly insightful, bridging optimization and numerical methods in an elegant way. The introduction of a kernelizable dual formulation with out-of-sample extensions makes the approach applicable to modern machine learning problems. A notable contribution is the robust kernel PCA variant based on ℓ₁ reconstruction error minimization, which enhances robustness to outliers. The connection to iteratively reweighted least squares (IRLS) provides a well-founded optimization approach for this problem. By connecting PCA, DC optimization, and numerical linear algebra, the paper helps unify perspectives from different domains, making it valuable for researchers in optimization, linear algebra, and machine learning.

Weaknesses:
While the connection between DCA and simultaneous iteration is intriguing, the paper does not provide explicit convergence rates or conditions under which the proposed algorithms are guaranteed to converge to the global minimum. Additionaly, for the robust $l_1$ PCA variant, the relationship to IRLS is mentioned, but formal convergence properties are not rigorously established.

---

> ### Author Rebuttal · Authors · 2025-07-30
>
> We thank the reviewer for their high evaluation, insightful questions and useful recommendations to improve our work.
>
> ---
>
> In the following we answer the posed **Questions**.
>
> > Proposition 3.3 appears to rely on the distinctness of singular values. Could this assumption be relaxed while still preserving the validity of the proposition? Alternatively, does the non-distinct case lead to fundamentally different results?
>
> The distinctness of singular values can be relaxed. For the proof of Proposition 3.3, it suffices to change scalars to scalar matrices, i.e. $\sigma_i$ becomes $\sigma_i I_{n_i}$, where $n_i$ denotes the multiplicity. It follows that Proposition 3.3 stays largely the same, with the only difference being that multiplicities should be taken into account and columns of $V$ may be permuted within the space of right singular vectors corresponding to the same singular value. We will remove this assumption in the revised manuscript.
>
>  We note that if this assumption does not hold, then the performance of classical methods degrades considerably (recall that the standard power iteration converges linearly with rate $\lambda_2/\lambda_1$ where the eigenvalues $\lambda_i$ of a symmetric matrix are sorted in decreasing order), see for example Table 7 where we consider spectra with small spectral gaps. Empirically, we therefore observe that the first-order methods on these formulations are less sensitive to the spectral properties. However, we would also like to note that this case does not occur in practice.
>
> > The paper introduces several novel optimization-based algorithms for PCA. Could you provide a more rigorous discussion of their convergence properties? Specifically: Under what conditions (e.g., step sizes, initialization) do they converge? Are there local vs. global convergence guarantees? How does their convergence rate compare to classical methods? For the robust ℓ₁ variant, does the connection to IRLS imply any inherited convergence guarantees?
>
> See our response to **Reviewer FsFQ** for convergence guarantees for the DCA algorithms. Regarding the convergence of PG and zeroFPR, the standard assumptions to minimize objectives of the form $f(x) + g(x)$ using these methods require $f$ to be $L$-smooth, while $g$ is often the indicator of a closed and convex set as in our case. In general, the (global) rates of PG and zeroFPR to stationary points are both sublinear $O(1/k)$ [1]. We note that, strictly speaking, for some formulations our $f$ is nonsmooth. However, this does not pose a problem since the objectives are concave. Therefore, the Euclidean descent lemma [2, Lemma 5.7] holds with $L=0$, and both PG and zeroFPR enjoy the same theoretical guarantees as described above. In fact, we may choose arbitrarily large stepsizes, despite nonsmoothness. This discussion will be added in the revised manuscript.
>
> The general rate for IRLS algorithms is also sublinear. However, under the restricted isometry property, it is possible to show a linear rate for IRLS [3]. While we have shown a connection between Algorithm 9 (DCA) and Algorithm 10 (IRLS), it is not immediately clear how to concretely map the iterates to each other so a linear rate in this setting is not immediately inherited. We conjecture that such a rate is also possible for our Algorithm 9 under the same property, though we believe that such an endeavor deserves separate work.
>
> > While Assumption 3.1 appears natural from a data processing standpoint, it represents a significant restriction from the perspective of numerical linear algebra where such constraints aren't typically required. If we relax this assumption, which of the proposed PCA reformulations remain valid?
>
> Assumption 3.1 is nowhere used in any of the derivations of alternate formulations, so all formulations remain valid. We have merely included this assumption so that the statistical interpretation of PCA is still valid (i.e. variance maximization). We will remove this assumption and only mention it when speaking about the statistical interpretation to make it clear that this is not a restriction of the paper.
>
> ---
>
> References:
>
> [1] Themelis, Andreas, Lorenzo Stella, and Panagiotis Patrinos. Forward-backward envelope for the sum of two nonconvex functions: Further properties and nonmonotone linesearch algorithms. SIAM Journal on Optimization 28(3) (2018): 2274-2303.
>
> [2] Beck, Amir. First-order methods in optimization. Society for Industrial and Applied Mathematics, (2017).
>
> [3] Daubechies, Ingrid, et al. Iteratively reweighted least squares minimization for sparse recovery. Communications on Pure and Applied Mathematics: A Journal Issued by the Courant Institute of Mathematical Sciences 63(1) (2010): 1-38.

---

> > ### Comment · Reviewer_jssA · 2025-08-08
> >
> > Thank you, everything is clear, and I continue to support the paper.

---

### Official Review · Reviewer_FsFQ · 2025-07-02

**Clarity:** 3
**Significance:** 3
**Originality:** 3
**Rating:** 4
**Confidence:** 5

**Summary:**

This paper revisits principal component analysis (PCA) through the lens of difference-of-convex (DC) duality, proposing several formulations, interpreting classical iterative methods such as simultaneous iteration as instances of DC, and introducing kernelizable formulations for robust variants of PCA.

**Questions:**

1. Include convergence guarantees or at least local convergence conditions for each of the proposed DCAs.
2. Benchmark against methods for sparse/robust PCA on real datasets (e.g., MNIST, Yale faces, motion segmentation).

**Ethical Concerns:**

["NO or VERY MINOR ethics concerns only"]

**Final Justification:**

Thanks, authors for the detailed the explanation. The referee is generally happy and positive about this paper. However, the referee feels it requires another round of review to check the correctness and proof details of the claims made. And, the convergence to the critical point is not satisfactory, since we know PCA can be polynomially solved to optimality. The authors' explanation of sparse PCA is interesting and should be incorporated into the manuscript. Again, referee is positive about this paper and retains the same score for this paper.

**Limitations:**

Yes.

**Paper Formatting Concerns:**

NA.

**Quality:**

3

**Strengths And Weaknesses:**

Strengths:
1. The paper presents a theoretical reinterpretation of PCA via DC duality and identifies connections with classical linear algebra routines.
2. The use of Toland duality is methodologically sound and allows formulation of new primal-dual pairs.
3. The kernelization of DC duals via unitarily invariant primal functions is a useful insight.

Weaknesses:
1. This paper fails to provide any theoretical convergence guarantees for DC algorithms. No rate of convergence, global optimality conditions, or even local convergence analyses are presented. Given the non-convex nature of the problems, such results are crucial for justifying the use of DCA in practice.
2. The empirical study does not convincingly demonstrate the superiority of the proposed DC methods over classical PCA solvers. In Table 1, DCA-based algorithms (Algorithms 1–6) often take longer than standard eigensolvers, such as SVDS and KrylovKit, with no clear accuracy benefit. The best-performing formulations are still dominated by classical methods in terms of both speed and simplicity, which weakens the practical motivation for these DC-based approaches.
3. The numerical experiments are limited to synthetic Gaussian data with a fixed dimensionality and target rank. There is no demonstration on real-world datasets, and no evaluation of approximation quality, numerical stability, or robustness to noise.

---

> ### Author Rebuttal · Authors · 2025-07-31
>
> We thank the reviewer for their positive feedback and useful comments to improve our work.
>
> ---
>
> In the following we address the stated **Weaknesses**.
>
> > This paper fails to provide any theoretical convergence guarantees for DC algorithms. No rate of convergence, global optimality conditions, or even local convergence analyses are presented. Given the non-convex nature of the problems, such results are crucial for justifying the use of DCA in practice.
>
> We did not provide any theoretical convergence guarantees for DCA due to space restrictions and the fact that it is a well-established algorithm whose theoretical basis is described in full detail in the literature, see for example [1]. Upon acceptance of the paper, we will make the paper more self-contained and further enhance the theoretical aspects by adding the following paragraph in the DCA section 2.3 regarding the basic convergence properties of DCA in general.
>
> Consider the problem (P) and suppose that the iterates $\{W^{(k)}\}$ are generated by (DCA), then
> - $G(W^{(k+1)}) - F(XW^{(k+1)}) \leq G(W^{(k)}) - F(XW^{(k)})$, i.e., (DCA) is a descent method (without linesearch). This fact can be seen from the fact that (DCA) fits into the majorization-minimization framework [2].
> - If $G(W^{(k+1)}) - F(XW^{(k+1)}) = G(W^{(k)}) - F(XW^{(k)})$, then (DCA) terminates at the $k$th iteration and $W^{(k)}$ is a critical point of (P). Here, a critical point $\bar{W}$ of (P) is defined as a point satisfying $\partial G(\bar{W}) \cap X^\top \partial F(X\bar{W}) \neq \emptyset$, which is a necessary condition for optimality. If $G(W) - F(XW)$ is bounded below, then every limit point of the sequence $\{W^{(k)}\}$ is a critical point. We note that the optimal value being bounded below is an assumption that is trivially satisfied for all our PCA formulations since the classical formulation optimizes a continuous function over a compact set, and the optimal values of the other formulations can be related through strong duality or problem transformations.
> - In general, the rate for (DCA) is sublinear (also known as linear in the complexity theory literature). However, if certain growth conditions hold, then linear rates are possible, see for example [3].
>
> Moreover, from the described connection between some of our algorithms and simultaneous iteration (Theorem 3.12), we inherit the linear convergence rate of simultaneous iteration [4, Theorem 28.4]. This insight will be added in the paragraph following Theorem 3.12.
>
> > The empirical study does not convincingly demonstrate the superiority of the proposed DC methods over classical PCA solvers. In Table 1, DCA-based algorithms (Algorithms 1–6) often take longer than standard eigensolvers, such as SVDS and KrylovKit, with no clear accuracy benefit. The best-performing formulations are still dominated by classical methods in terms of both speed and simplicity, which weakens the practical motivation for these DC-based approaches.
>
> It is true that some of the DC methods do not perform better than the classical PCA solvers. This is to be expected since we discovered that these are related to simultaneous iteration, which is the default eigenvalue solver for finding *complete* eigendecompositions (the QR algorithm or more generally QZ algorithm). Since the setting we consider only cares about the top principal components, the specialized algorithms which extract few components are naturally better. We have merely included these algorithms as a baseline comparison and to show their (unexpected) connection to numerical linear algebra. These methods can be thought of as the “gradient descent” for DC problems and there are better algorithms that can be derived, such as ADMM-type algorithms [5] as well as those inspired by the connection between (DCA) and Frank-Wolfe [6].
>
> We respectfully disagree regarding the assessment that the best-performing formulations are dominated by the standard eigensolvers in both speed and simplicity. As for speed, both Table 1 and the additional experiments in the Appendix show that ZeroFPR and PG perform better than the classical methods for a certain range of required accuracies. Regarding the simplicity, we wish to emphasize that we compare generic first-order methods against tailored state-of-the-art implementations that include many low-level optimizations. Hence the main takeaway we hope to convey through our preliminary experiments is that for the level of accuracy typically sufficient in machine learning applications, there exist formulations for which faster solvers can be designed.
>
> > The numerical experiments are limited to synthetic Gaussian data with a fixed dimensionality and target rank. There is no demonstration on real-world datasets, and no evaluation of approximation quality, numerical stability, or robustness to noise.
>
> We would like to emphasize that the main contribution of this paper is its theory, which all the reviewers have acknowledged to be sound. In that sense, we believe that a thorough empirical study should be deferred to a separate paper. Moreover, our experiments show that generic first-order optimization methods can beat state-of-the-art numerical linear algebra routines in the “machine learning regime”, which is an unexpected observation that goes against conventional wisdom. Regarding numerical stability, since our formulations are based on Schatten-norms, their implementations require (small) SVDs, which operate through unitary matrices and are therefore numerically stable. See our response to **Reviewer BYYV** for additional experiments.
>
> ---
>
> In the following we answer the posed **Questions**.
>
> > Include convergence guarantees or at least local convergence conditions for each of the proposed DCAs.
>
> See our response to the first **Weakness** as well as our response to the second question of **Reviewer jssA**.
>
> > Benchmark against methods for sparse/robust PCA on real datasets (e.g., MNIST, Yale faces, motion segmentation).
>
> We would like to point out that classical robust PCA formulations [7] are of a different nature than our proposed method. The classical methods often decompose a matrix into a low-rank component and a sparse component, which are well-suited for motion segmentation. However, our robust formulation views PCA from its autoencoder perspective and aims to enforce sparsity on the reconstruction error, which is better aligned with outlier detection (note that this is also different from sparse PCA). Another major difference is that most classical methods are not out-of-sample applicable while ours is through Theorem 3.9. These differences make fair comparisons difficult.
>
> To further illustrate this fact, consider the synthetic experiment from [7, Section 4.1] (note that they do not present any other numerical results with metrics that can be easily compared). We first generate a low-rank component $L^\star$ according to the same settings and add a sparse matrix $E$ where $E$ has limited support and the entries of $E$ are independent Bernoulli $\pm 1$ entries. We consider two settings:
> 1. the support of $E$ is chosen uniformly distributed over all its entries.
> 2. the support of $E$ consists exclusively of full rows (i.e., an image where complete rows are perturbed).
>
> We then compare the classical principal component pursuit (PCP) with alternating directions [7, Algorithm 1] and our Algorithm 5. For our experiment, we choose $\\mathrm{rank}(L^\star) = 25$ and $L^\star\in\mathbb{R}^{500\times 500}$. We assume the size of the support of $E$ is $|E| = 12500$ and compute both the reconstruction error of the low-rank component and the primal cost of Proposition 4.1 (i.e., the $l_1$-norm of the row-wise reconstruction errors) and summarize our result in the following table (all metrics were averaged over $20$ different random problems).
>
> |           | $\\\|L - L^\\star\\\|_F / \\\|L^\\star\\\|_F$ PCP | $\\\|L - L^\\star\\\|_F / \\\|L^\\star\\\|_F$ Alg 5 | Primal cost PCP | Primal cost Alg 5 |
> | --------- | ------------------------------------------------- | --------------------------------------------------- | ------------------------ | -------------------------- |
> | Setting 1 | $3 \\times 10^{-8}$                               | 0.013                                               | 2488                     | 2364                      |
> | Setting 2 | 0.045                                             | 0.009                                               | 559                      | 545                      |
>
>
> We observe that the classical algorithm from [7] performs better in the setting where it was proposed while our algorithm is better in the outlier setting. Moreover, we see that the primal cost function is lower for our algorithm in both cases, which is to be expected since the PCP algorithm is not designed to minimize this cost.
>
> ---
>
> References
>
> [1] Tao, Pham Dinh, and LT Hoai An. Convex analysis approach to DC programming: theory, algorithms and applications. Acta mathematica vietnamica 22(1) (1997): 289-355.
>
> [2] Sun, Ying, Prabhu Babu, and Daniel P. Palomar. Majorization-minimization algorithms in signal processing, communications, and machine learning. IEEE Transactions on Signal Processing 65(3) (2016): 794-816.
>
> [3] LT, Hoai An, Van Ngai Huynh, and Pham Dinh Tao. Convergence analysis of difference-of-convex algorithm with subanalytic data. Journal of Optimization Theory and Applications 179(1) (2018): 103-126.
>
> [4] Trefethen, Lloyd N., and David Bau. Numerical linear algebra. Society for Industrial and Applied Mathematics, (2022).
>
> [5] Sun, Tao, et al. Alternating direction method of multipliers with difference of convex functions. Advances in Computational Mathematics 44(3) (2018): 723-744.
>
> [6] Yurtsever, Alp, and Suvrit Sra. CCCP is Frank-Wolfe in disguise. Advances in Neural Information Processing Systems 35 (2022): 35352-35364.
>
> [7] Candès, Emmanuel J., et al. Robust principal component analysis?. Journal of the ACM (JACM) 58(3) (2011): 1-37.

---

> > ### Comment · Reviewer_FsFQ · 2025-08-05
> >
> > The referee appreciates the authors' efforts. The concerns summarized in the weakness section are still valid. Especially, no new theoretical guarantees for the algorithms and it only provides a new perspective. And this method, in referee's opinion, cannot address sparsity at this stage. Hence, the contributions are less significant. Nevertheless, the referee retains the same positive score for this manuscript.

---

> > > ### Author Response · Authors · 2025-08-06
> > >
> > > Dear Reviewer FsFQ
> > >
> > > Thank you for your comment and positive evaluation of our work.
> > >
> > > Could you clarify which weaknesses have not been addressed adequately, so we can improve this in the manuscript? We will add the convergence rates as well as the optimality conditions. We emphasize that we propose novel algorithms in the sense of applying existing (meta) algorithms to novel formulations. This is for example also the case in [1], one of the original papers to propose ADMM for principal component pursuit, where they also do not provide new theoretical guarantees since there already exists a rich literature. Further, our preliminary experiments show that there exist formulations for which algorithms can be designed to outperform state-of-the-art methods. Lastly, we have done both standard PCA and robust PCA experiments on real-world datasets, as requested, as well as compared our robust PCA with a classical PCA formulation in their setting.
> > >
> > > Regarding sparse PCA (which is different from robust PCA), since this was not one of the goals of this paper and also not stated as a weakness in the initial review, we have not discussed it.
> > >
> > > Nevertheless, there are several DC formulations for sparse PCA.
> > > 1) Inspired by formulation (q), one can propose the novel DC formulation $\mathrm{minimize}\_{W\in\mathbb{R}^{d\times s}} \frac{1}{2} \\|W\\|\_S^2 - \\|XW\\|\_{S\_2}$
> > > where $\\|\cdot\\|\_S$ is a sparsity-inducing norm such as taking the sum of the absolute value of its elements ($l\_1$-norm on its vectorization). Note that is not a classical sparse PCA formulation but achieves a similar purpose. The convex conjugate of this function follows from (1) in the Appendix.
> > > 2) The standard SDP formulation for SPCA (see [1]) can be written in DC form as $\mathrm{minimize}\_{W\in\mathbb{S}^{d\times d}} \delta\_C(W) - \frac{1}{2}\\|XW\\|\_{S\_2}^2$ where $\delta\_C(W)$ is the indicator of a convex set $C$ such that $W$ is sparse. For the SDP formulation, $C$ is given by $\\{W \in \mathbb{S}^{d\times d} \mid \mathrm{trace}(W)=1, \sum\_{i,j=1}^n |X\_{ij}| \leq k, X \succeq 0\\}$. The convex conjugate of this indicator is the support function of $C$, though it is not clear whether this admits a closed form expression.
> > > 3) If we consider only the first principal component (this process can then be repeated via deflation), sparse PCA has the DC formulation $\mathrm{minimize}\_{w\in\mathbb{R}^d} \delta\_C(w) - \frac{1}{2}\\|Xw\\|\_{S\_2}^2$, where $C$ is the convex hull of the intersection of the unit norm ball and $\\{w\in\mathbb{R}^d \mid \\|w\\|\_0 \leq k\\}$. We have once again relaxed the constraint set to its convex hull since this formulation maximizes a convex function, as we have also done in the paper. The convex conjugate of this indicator can be computed in closed form. To see this, note that $\delta^\*\_C(z) = \max\_{w \in C} w^\top z$. Since $w$ has at most $k$ nonzeros, we see that this maximization problem is solved when the support of $w$ corresponds to the $k$ largest entries of $z$ in absolute value. Taking into account that $w$ is also a unit vector, it follows that $\delta^\*\_C(z)$ is exactly the Euclidean norm of the vector containing the $k$ largest entries of $z$ in absolute value.
> > >
> > > These formulations are not unitarily invariant and therefore not easily kernelizable through Proposition 3.6 (a sufficient condition for kernelizability). Though, to the best of our knowledge, sparse PCA is generally not used in the kernel setting since the main motivation for sparse PCA is interpretability of the data and principal components. More concretely, by restricting the cardinality of the principal components, this implicitly assumes that they are linear combinations of just a few input variables, which does not mesh well with kernel methods that are inherently nonlinear.
> > >
> > > ---
> > >
> > > We would also like to highlight the experiment in our response to **Reviewer BYYV** where we show how our robust kernel PCA formulation on a contaminated dataset can achieve performance on par with standard methods on the noncontaminated data, which shows that the formulation is able to extract relevant features despite many outliers. Moreover, this out-of-sample-applicability is not possible for classical robust PCA, let alone in the kernel setting.
> > >
> > > ---
> > > References
> > >
> > > [1] Yuan, Xiaoming, and Junfeng Yang. Sparse and low-rank matrix decomposition via alternating direction methods. (2009).
> > >
> > > [2] d'Aspremont, Alexandre, et al. A direct formulation for sparse PCA using semidefinite programming. Advances in neural information processing systems 17 (2004).

---

> > > > ### Comment · Reviewer_FsFQ · 2025-08-09
> > > >
> > > > Thank you, authors for the detailed the explanation. The referee is generally happy and positive about this paper. However, the referee feels it requires another round of review to check the correctness and proof details of the claims made. And, the convergence to the critical point is not satisfactory, since we know PCA can be polynomially solved to optimality. The explanation of sparse PCA is also interesting and should be incorporated into the manuscript. Again, referee is positive about this paper and retains the same score for this paper.

---

### Official Review · Reviewer_BYYV · 2025-07-03

**Clarity:** 3
**Significance:** 2
**Originality:** 2
**Rating:** 4
**Confidence:** 2

**Summary:**

This paper revisits Principal Component Analysis (PCA) through the lens of difference-of-convex (DC) duality, particularly Toland duality, and presents several contributions:
1. New DC Dual Pairs for PCA: The authors derive three new dual formulations of PCA and show that classical algorithms (like simultaneous iteration and the QR algorithm) can be interpreted as instances of the Difference-of-Convex Algorithm (DCA).
2. Robust and Kernelizable PCA: A new DC formulation is proposed for robust PCA, including a dual that is kernelizable and supports out-of-sample extension, which is significant for real-world applications.
3. Efficient Algorithms: Several DCA-based algorithms are proposed and compared empirically against classical and modern methods, showing strong performance, especially with newly introduced formulations.

**Questions:**

1. Have you tried your robust kernel PCA formulation on any real-world datasets with known outliers? How does it compare to classic robust PCA variants?
2. You mention that standard kernel PCA methods suffer from scalability issues. Have you evaluated the performance of your kernelizable DC duals with Nyström approximation or other scalable kernel methods?
3. You report that formulations (n)-(o) perform best in your experiments. Do you have any theoretical insights into why these perform better than others (e.g., (l)-(m))? Is it due to smoother optimization landscapes or better conditioning?
4. You discuss the link between PCA and linear autoencoders. Could the DC duality framework also inspire new architectures or training methods for nonlinear or deep autoencoders?

**Ethical Concerns:**

["NO or VERY MINOR ethics concerns only"]

**Final Justification:**

I appreciate the effort the author put into addressing my concerns, and I’m satisfied with the explanation made by the authors.

**Limitations:**

Yes

**Quality:**

2

**Strengths And Weaknesses:**

Strengths:

The reinterpretation of PCA through DC duality, and the discovery of new dual pairs (beyond existing literature), are mathematically insightful. Showing that classical methods like simultaneous iteration are instances of DCA gives new understanding into optimization perspectives in numerical linear algebra. The kernelizable robust PCA formulation is a practical strength that aligns well with modern machine learning use cases.

Weaknesses:

While the theoretical results are strong, it's unclear how well the proposed methods extend to large-scale, high-noise, or real-world datasets. The experiments are mostly synthetic or based on standard PCA settings. A demonstration on real-world tasks (e.g., image compression, NLP embeddings) would be helpful.

---

> ### Author Rebuttal · Authors · 2025-07-31
>
> We thank the reviewer for their positive evaluation of our work and interesting questions.
>
> ---
>
> In the following we address the **Weakness** stated by the reviewer.
>
> > While the theoretical results are strong, it's unclear how well the proposed methods extend to large-scale, high-noise, or real-world datasets. The experiments are mostly synthetic or based on standard PCA settings. A demonstration on real-world tasks (e.g., image compression, NLP embeddings) would be helpful.
>
> We will add the following experimental results to the manuscript. Consider the following 2 datasets.
> - MNIST dataset (60000x784)
> - The 100k top words from the 2024 Wikipedia + gigaword 5, 50d GloVe word embeddings dataset (100000x50)
>
> We compare the computational times of classical PCA eigensolvers against our methods with a required tolerance of $\varepsilon = 10^{-3}$. The results for the best algorithms are summarized in the following table where the column header denotes the number of principal components (no standard deviation is provided for timings larger than 5 seconds).
>
> MNIST:
> |             | 10                       | 30                         | 50                       | 100                        | 150                        |
> | ----------- | ------------------------ | -------------------------- | ------------------------ | -------------------------- | -------------------------- |
> | ZeroFPR (n) | $125.880 \pm 28.716$ ms | $286.692 \pm 92.200$ ms   | $444.745 \pm 86.127$ ms | $1734.389 \pm 247.000$ ms | $2363.292 \pm 135.235$ ms |
> | PG (n)      | $111.101 \pm 9.363$ ms  | $215.460 \pm 35.660$ ms   | $351.948 \pm 62.360$ ms | $1254.828 \pm 103.131$ ms  | $1772.221 \pm 150.473$ ms |
> | SVDS        | $820.029 \pm 85.984$ ms | $1685.063 \pm 226.155$ ms | $2576.462 \pm 4.456$ ms | $5251.069$ ms                | $7850.230$ ms                |
>
> GloVe:
> |             | 5                        | 10                         | 15                         | 20                         | 30                         |
> | ----------- | ------------------------ | -------------------------- | -------------------------- | -------------------------- | -------------------------- |
> | ZeroFPR (n) | $61.963 \pm 79.404$ ms  | $131.065 \pm 114.521$ ms | $208.227 \pm 150.357$ ms | $271.817 \pm 147.240$ ms | $367.117 \pm 198.195$ ms |
> | PG (n)      | $43.312 \pm 59.676$ ms  | $75.954 \pm 71.368$ ms   | $119.950 \pm 100.352$ ms | $165.287 \pm 111.865$ ms | $250.385 \pm 113.187$ ms |
> | SVDS        | $136.180 \pm 19.130$ ms | $200.576 \pm 14.643$ ms  | $281.008 \pm 31.438$ ms  | $319.668 \pm 11.535$ ms  | $328.987 \pm 34.468$ ms  |
>
> In both of the experiments, we observe the same main observations as from the synthetic experiments: the generic first-order methods are faster than the classical eigensolvers in this setting. However, we also remark that the difference for the GloVe dataset is quite small. We believe this is the case because the word embeddings have favorable spectral properties such that the results are quite similar to the first column of Table 7.
>
> ---
>
> In the following we answer the posed **Questions**.
>
> > Have you tried your robust kernel PCA formulation on any real-world datasets with known outliers? How does it compare to classic robust PCA variants?
>
> To demonstrate the robust feature extraction of the robust PCA formulation, consider the MNIST dataset (train-test split 80-20) and contaminate 15% of the training data with heavy Gaussian noise. We then consider the following settings:
> 1. linear PCA on the noncontaminated data (baseline)
> 2. linear PCA on the contaminated data
> 3. robust PCA on the contaminated data using Algorithm 5
> 4. robust PCA on the contaminated data using Algorithm 6 with a linear kernel (note that the kernel matrix does not fit in memory but we can use its factored representation $XX^\top$ since the kernel is linear)
>
> For each of these settings, we evaluate the reconstruction error on the test set, which is not contaminated! These errors are summarized in the following table (the column headers denote the number of principal components).
> |           | 50    | 100    | 150    |
> | --------- | ----- | ------ | ------ |
> | Setting 1 (noncontaminated) | 0.011 | 0.0057 | 0.0034 |
> | Setting 2 (contaminated)  | 0.062 | 0.0569 | 0.0520 |
> | Setting 3 (contaminated)  | 0.014 | 0.0107 | 0.0090 |
> | Setting 4 (contaminated)  | 0.014 | 0.0107 | 0.0090 |
>
> Comparing Settings 3 and 4 with Setting 2, we observe that our robust formulation better reconstructs the test set when trained on the contaminated data. The fact that Setting 3 and 4 have similar performance is expected from strong duality. Moreover, the top components are less affected by the outliers than the remaining components. This is logical since these latter components explain less “variance” and it becomes more difficult to distinguish noise/outliers from data.
>
> To further illustrate the strength of the formulation, we can now extract robust features in light of Theorem 3.9. We train a small MLP classifier (1 hidden layer with 20 neurons) on these extracted features (we choose a very simple classifier so the quality of the features becomes more apparent). Additionally, we also consider the following two settings:
>
> 5. robust PCA on the noncontaminated data using Algorithm 5
> 6. robust kernel PCA on the contaminated data using Algorithm 6 with a RBF kernel ($\gamma=0.01$) approximated using a Nyström approximation with 500 pivots
>
> We summarize the test accuracies in the following table (the column headers indicate number of principal components at the intermediate stage)
>
> |                             | 50    | 100   | 150   |
> | --------------------------- | ----- | ----- | ----- |
> | Setting 1 (noncontaminated) | 95.87 | 95.73 | 94.84 |
> | Setting 2 (contaminated)    | 76.57 | 79.93 | 81.09 |
> | Setting 3 (contaminated)    | 86.89 | 87.32 | 85.53 |
> | Setting 4 (contaminated)    | 87.63 | 84.44 | 86.92 |
> | Setting 5 (noncontaminated) | 95.73 | 96.06 | 95.43 |
> | Setting 6 (contaminated)    | 95.21 | 95.43 | 94.34 |
>
> We observe:
> - The robust formulations always perform better than the non-robust formulation for the contaminated data.
> - Using a robust formulation (Setting 5) on the clean data does not degrade performance with respect to the standard formulation (Setting 1).
> - The RBF kernel trained on the corrupted data performs on par with models trained on the clean data.
>
> For a comparison with a classical robust PCA formulation (that is not out-of-sample applicable) on a synthetic experiment, see our response to **Reviewer FsFQ**. All the code will be made public later.
>
> > You mention that standard kernel PCA methods suffer from scalability issues. Have you evaluated the performance of your kernelizable DC duals with Nyström approximation or other scalable kernel methods?
>
> See our answer to the previous question (a Nyström approximation is required for the MNIST dataset since storing the complete kernel matrix of the training data would take 20+ GB of memory).
>
> > You report that formulations (n)-(o) perform best in your experiments. Do you have any theoretical insights into why these perform better than others (e.g., (l)-(m))? Is it due to smoother optimization landscapes or better conditioning?
>
> We indeed believe that the formulations (n)-(o) have a better condition number than the formulations (l)-(m), leading to their better performance. We’d like to point out an analogy with solving linear least squares where it is known that solving the normal equations is impractical, as its condition number is squared, which is similar to formulations (l)-(m).
>
> > You discuss the link between PCA and linear autoencoders. Could the DC duality framework also inspire new architectures or training methods for nonlinear or deep autoencoders?
>
> As briefly noted in the introduction, the transformer architecture bears a close connection to kernel PCA, which has already motivated the development of a robust transformer variant in [1]. Similarly, an alternative transformer design was proposed in [2], based on another duality framework. Deep autoencoders derived from PCA-based formulations are also identified in our future work, and related architectures based on alternative formulations have been explored as well, see [3]. We therefore indeed believe that the DC duality framework can inspire new architectures as well as efficient training methods.
>
> ---
>
> References
>
> [1] Rachel SY Teo and Tan Nguyen. Unveiling the hidden structure of self-attention via kernel principal component analysis. Advances in Neural Information Processing Systems, 37:101393–101427, 2024.
>
> [2] Yingyi Chen, et al. Primal-attention: Self-attention through asymmetric kernel svd in primal representation. Advances in Neural Information Processing Systems, 36:65088–65101, 2023.
>
> [3] Tonin, Francesco, et al. Deep Kernel Principal Component Analysis for multi-level feature learning. Neural Networks 170 (2024): 578-595.

---

> > ### Comment · Reviewer_BYYV · 2025-08-08
> > **Reply**
> >
> > Thank you for your very thorough and considerate reply. I appreciate the effort you put into addressing my concerns, and I’m satisfied with your explanation.

---

### Note · Authors · 2025-08-13

Dear Area Chairs and Reviewers,

We would like to thank you for your time and valuable comments, which have helped us improve our work.

In this work, we have presented multiple new and useful difference-of-convex formulations for PCA, and found novel connections between PCA, optimization and numerical linear algebra. Moreover, we described a family of PCA-like formulations that are kernelizable and out-of-sample applicable, which can inspire many more formulations for various applications. In particular, we investigated a robust kernelizable PCA based on an $l_1$-minimization of the reconstruction errors. Lastly, some preliminary experiments show that based on these novel formulations, generic first-order methods can beat state-of-the-art solvers for the accuracy that is required in machine learning applications.

The reviewers agree that our approach is methodologically sound and presents useful insights.

After the rebuttal, we have taken the reviews into account and incorporated the following changes in the manuscript:
- We have included additional experiments on real-world datasets, the robust PCA formulation and comparisons with classical robust PCA (see replies to **Reviewer BYYV** and **Reviewer FsFQ**).
- We have added the standard theoretical guarantees for DCA, PG and ZeroFPR. Moreover, as a corollary of the connection between some of our DCA algorithms and simultaneous iteration, we inherit the linear rate as well as the convergence to **global optimality** with probability $1$, starting from random initialization. Similarly, since PG for minimizing $-F$ has the well-known interpretation of DCA applied to $(G + (1/2)\\|\cdot\\|^2) - (F + (1/2)\\|\cdot\\|^2)$, where $G$ denotes the indicator of a convex set, these can be connected to simultaneous iteration on an identity-shifted covariance/kernel matrix (i.e., it naturally incorporates regularization), such that again convergence to **global optima** is guaranteed.
- We have relaxed some assumptions, as noted by **Reviewer jssA**.
- Wherever possible, we have improved the writing of the text for enhanced readability and better highlighted our contributions and their implications. Minor typographical changes, like those suggested by **Reviewer A5aE**, have also been made.
- A short discussion on DC formulations for sparse PCA has been added.

---

### Decision · Program_Chairs · 2025-09-17

**Decision:**

Accept (poster)

**Comment:**

The submission studies the classical PCA problem from the perspective of difference-of-convex (DC) optimization. While DC optimization is known to be generally intractable, some heuristics tend to perform fairly well on practical instances. This is a situation that resembles some numerical lineal algebra problems, such as nonnegative matrix factorization or sparse PCA. Hence, I find this approach natural and insightful.

Reviews agree on the technical merits of this work, but some have concerns about the experimental evaluations. Authors provided further results in this line; while I find this not strictly necessary, it is appreciated that the authors provide more thorough evaluations.

Based primarily on the technical contributions, this work seems suitable for publication.